# Mapping snow depth over lake ice in Canada's sub-arctic using ground-penetrating radar

Alicia F. Pouw, Homa Kheyrollah Pour, Alex MacLean,

Remote Sensing of Environmental Change (ReSEC) Research Group, Department of Geography and Environmental Studies,

Wilfrid Laurier University, Waterloo, N2L 3C5, Canada

Cold regions Research Centre, Wilfrid Laurier University, Waterloo, N2L 3C5, Canada

*Correspondence to*: Alicia F. Pouw (apouw@wlu.ca)

## Abstract

Ice thickness across lake ice is influenced mainly by the presence of snow and its distribution, which affects the rate of lake

ice growth. The distribution of snow depth over lake ice varies due to wind redistribution and snowpack metamorphism, affecting the variability of lake ice thickness. Accurate and consistent snow depth data on lake ice are sparse and challenging to obtain. However, high spatial resolution lake snow depth observations are necessary for the next generation of thermodynamic lake ice models to improve the understanding of how the varying distribution of snow depth influences lake ice formation and growth. This study was conducted using ground-penetrating radar (GPR) acquisitions with ~9 cm sampling

resolution along transects totaling ~44 km, to map snow depth over four freshwater lakes in Canada's sub-arctic. The lake snow depth derived from GPR TWT resulted in an average relative error of under 10% when compared to 2,430 in situ snow depth observations for early and late winter season. The snow depth derived from GPR TWTs for the early winter season was estimated with a root mean square error (RMSE) of 1.6 cm and a mean bias error of 0.01 cm, while the accuracy for the late winter season on a deeper snowpack was estimated with a RMSE of 2.9 cm and a mean bias error of 0.4 cm. The GPR-derived

snow depths were interpolated to create 1 m spatial resolution snow depth maps. The findings showed improved lake snow depth retrieval accuracy and introduced a fast and efficient method to obtain high spatial resolution snow depth information. The results suggest that GPR acquisitions can be used to derive lake snow depth, providing a viable alternative to manual snow depth monitoring methods. The findings can lead to an improved understanding of snow and lake ice interactions, which is essential for northern communities' safety and wellbeing and the scientific modelling community.

## 1 Introduction

The distribution of snow depth over lake ice affects the formation and thickness of ice over the entire lake. While snowfall can accelerate the onset of lake freeze-up, once the ice has formed, the accumulation of snow hinders the ice growth in the water column (Adams, 1976a). Snow present on top of lake ice acts as an insulative barrier due to its lower thermal conductivity

compared to ice. This process slows the growth rate of congelation ice (or black ice; Brown and Duguay, 2010; Leppäranta, 2015) and affects the heat released from the water column to the atmosphere. However, snow on lake ice can also impact the timing of melt and the ice-free season. The albedo of the snow surface reflects incoming solar radiation and can lead to a longer ice-on season (Jensen et al., 2007; Brown and Duguay, 2011; Robinson et al., 2021). Moreover, snow can produce ice growth as snow ice (or white ice), if the snow on the ice surface encounters water, which forms slush and refreezes (Leppäranta, 1983). This process can occur through the upwelling of water through leads, precipitation falling as rain, or heavy snow causing the depression of ice below the water level.

A challenge to measuring lake snow is the inconsistent snow thickness across the lake. Snow redistributed by wind commonly deposits on the leeward side of topographic features. Snow accumulation on lake ice surrounding these features (i.e., pressure ridges) leads to the formation of snowdrifts. Additionally, snow dunes will form in areas of turbulent winds on relatively level ice surfaces (Sturm and Liston, 2003; Liston et al., 2018). The formation of snowdrifts and snow dunes create a heterogenous snow thickness across the ice surface. The uneven snow depth distribution leads to spatial variability in the lake ice thickness due to the increase in heat transfer through the snow for areas of shallow snow (assuming a constant thermal conductivity). These micro-topographic snow features impact the ice mass balance and must be considered when evaluating the energy balance and fluxes on local and regional scales (Sturm et al., 2002).

Snow and lake ice are sensitive to a change in daily air temperature (Rafat et al., 2023). Northern Canada is experiencing warming at twice the global rate, and it is expected that air temperature will continue to increase, along with precipitation (about 10%) in all seasons (Zhang et al., 2019). These changes can significantly impact the surface-atmosphere energy balance which can directly affect snow and lake ice conditions (Brown and Duguay, 2010). As a result of these changes, alterations in snow cover (Brown et al., 2021; Mudryk et al., 2017), snowfall (Vincent et al. 2018), lake ice phenology (timing of ice formation and break-up; Magnuson et al., 2000; Lei et al., 2012; Benson et al., 2011), and ice thickness and composition (Kholoptsev et al., 2021) are being observed. Spatial and temporal observations of lake snow and ice can provide insights to changes in climatic variables. Later freeze up and earlier break-up of ice cover lead to an extended open-season, which can affect lake surface water temperatures (i.e., Woolway et al., 2021), influencing lake biogeochemical processes (e.g., Adrian et al., 2009; Jeppesen et al., 2014). Furthermore, northern communities rely on lake ice for cultural and recreational use, and as a source of transportation through ice roads (Knoll et al., 2019). Ice roads allow travel to neighbouring communities and alternative access to goods and supplies (instead of transport via airplane). With warming projected to increase, it can be expected that the safety of ice roads and operational duration will be affected (Stephenson et al., 2011; Mullan et al., 2021). As the presence of snow over lake ice directly affects ice thickness, measuring snow depth on lake ice is crucial for lake modelling and ice thickness estimation on a regional scale. A previous study by Kheyrollah et al. (2017) shows that accurate snow depth observations over lake ice can significantly improve thermodynamic lake ice models. Enhancing model input can lead to improved information on past, present, and future lake ice conditions to assist in climate change adaptation and decision making across Canada's North.

Improving snow depth observations and retrieving an accurate higher spatial resolution snow depth is essential for hydrological, limnological, and lake ice studies (Lei et al., 2012; Kheyrollah Pour et al., 2017; Marsh et al., 2020; Li et al., 2022). Daily snow depths are reported across Canada using instruments, such as a manual ruler or a sonic sensor, at weather

stations located on land (Brown et al., 2021). However, the depth of snow on land does not compare to snow over lake ice (Sturm & Liston, 2003). Snow depth over lake ice is ~ 30 % less than that over land (Gunn et al., 2015; Kheyrollah Pour et al., 2017), such that incorporating land-based snow observations into a thermodynamic lake ice model would negatively bias the ice thickness estimations. The distribution of snow over lake ice is affected more significantly by wind due to the open nature of lakes and the lack of vegetation catchments, which also create a heterogeneous snow surface across the lake ice (Adams,

1976a).

Currently, retrieving accurate lake snow depth observations and mapping the spatial distribution and heterogeneity of snow over ice is challenging because of the limited support of point measurements using contemporary methods, such as a ruler and notebook or automatic snow depth probe. An automatic snow depth probe, such as the magnaprobe, is equipped with a metal rod probe that penetrates the snowpack to the ice surface and a sliding basket that sits on the surface of the snow, recording

the snow depth and spatial location when manually placed in position (Sturm and Holmgren, 2018). The magnaprobe records the snow depth accuracy with errors ranging from near zero for hard bases to +5 cm. The Wide Area Augmentation System-enabled GPS provides a position accurate to ±2.5 m. An advantage of using a magnaprobe is the increase in speed with which a depth and position measurement can be obtained compared to measuring with a traditional ruler and writing down the results. The highest boost in snow depth measurement efficiency occurs when the distance between measuring locations is kept

relatively small. The snow depth probe has been commonly utilized for validation of remote sensing techniques (i.e., McGrath et al., 2019; Walker et al., 2020). However, due to the limited spatial coverage that the automated snow depth probe or ruler pose; it is not logistically feasible to measure the snow depth on lake-wide scales. Recent advancements have utilized Structure from Motion (SfM) from remotely piloted aircraft system (RPAS) acquisitions to map snow depth over land (i.e., Harder et al., 2016; 2020; Walker et al., 2020; King et al., 2022). This technique is limited in representing the lake ice surface elevation

because the ice surface is rarely exposed prior to snow accumulation, and the accumulation of snow, which submerges the ice, invalidates the elevation baseline (Adams, 1976b). A freeboard correction compensates for the change in ice surface elevation to the open water surface; however, this method requires prior information on the snowpack and ice thickness (Gunn et al., 2021a). Ground-penetrating radar (GPR) is one technique that can simultaneously estimate snow depth and ice thickness to be applied within the freeboard correction. GPR systems transmit an electromagnetic (EM) wave and record the measured

amplitude as a function of two-way travel-time (TWT) as the signal travels from the transmitting antenna, through a medium and reflects back to the receiving antenna at each interface. Although GPR is a recognized tool for measuring the spatio-temporal patterns of deep snow over land, sea ice, and glacial firn (i.e.,Webb, 2017; Webb et al., 2018; McGrath et al., 2019, 2022; Meehan et al., 2020; 2021; Pfaffhuber et al., 2017), it still requires observation of dry snow density (or snow depth and the radar travel-time for calibration) to derive snow depth from the GPR TWT (Marshall et al., 2005). Over lake ice, GPR is

commonly used to retrieve ice thickness (i.e., Barrette, 2011; Gunn et al., 2021a; 2021b); however, lake snow depth retrieval

using GPR is challenging due to the GPR signal attenuations, as well as the shallow snow-ice interface. In using GPR to derive lake ice thickness, the snow has commonly been ignored and best practices suggest avoiding ununiform snow (Sensors and Software, 2016). The snow causes thicker ice thickness estimates due to the radar travel-time increase, however, areas with thicker snowpacks are expected to have a shallower ice thickness. These challenges are mitigated through additional signal processing of the radargrams to identify the snow-ice interface and derive the shallow snow depth, as presented in this work. Our goal is to improve the knowledge and understanding of snow depth distribution over lake ice. We utilize extensive GPR two-way travel-time (TWT) observations and in situ observations of lake snow depth and density to complete the following objectives: (1) To improve the retrieval of lake-specific snow depth observations by applying a fully automated GPR processing algorithm, (2) To validate the accuracy of the snow depth retrieval algorithm by comparing it to in situ observations, and (3) To spatially map the distribution of snow depth across lakes. The outcome will increase lake snow depth data availability which benefits the hydrological and lake ice modelling communities.

**2 Study area**

In this study, GPR is used to derive and map snow depth over lake ice on four freshwater lakes located north of Yellowknife, NWT during the early and late winter season, such as Landing Lake (62.5587 °N, 114.4103 °W), Finger Lake (62.5750 °N, 114.3587 °W), Long Lake (62.4772 °N, 114.4422 °W), and Vee Lake (62.5555°N, 114.3502 °W) shown in Figure 1. All four lakes are located within the North Slave region. These lakes are generally covered by ice from October to April. The four lakes are close in proximity to one another but vary in shape and size (Table 1). It is expected that the wind fetch and shoreline vegetation affect the snow distribution on these lakes differently. This study uses data collected on areas within the four lakes, as identified in Figure 1b, covering regions along the shoreline, as well as open areas.

Data collection for this study took place during the 2021-2022 early winter season (between December 7th to 14th, 2021) for all lakes, as well as during the late season (March 27th, 2022) to capture the variability of snow depth in late season on a deeper snowpack on Landing Lake. Here, we will refer to Landing-D Lake to represent data collected in December and Landing-M Lake to represent data collected in March. The other three lakes will be referred to as Finger, Long and Vee Lakes. These lakes are part of a turbulent wind field, as the wind direction and speed reported at the Yellowknife weather station vary rapidly. The most predominant winds in December and November came from the east (~27%) and had an average wind speed of 2.5 m/s, with the strongest winds coming from the northeast (~15%) reaching 9 m/s. Throughout January to March, the strongest winds came from the northwest (~22%) reaching 10 m/s, but frequent winds came from the northeast in January (~ 22%), northwest in February (~26%) and northeast, east, and northwest in march (~21%) travelling at 3 m/s on average, while very little winds were recorded from the south (~6%) between October to March. During initial data collection, air temperatures ranged from -30°C to -15°C, and initial snow on the ground (December 7th, 2021) reported on land at the nearby Meteorological Service of Canada Yellowknife A weather station was 18 cm (Figure 2). During the time spent in the field, an additional 8 cm

of snow fell (December 7 to 14th, 2021). Returning in March 2022, the initial snow on the ground was reported at 42 cm and air temperatures around -20°C.

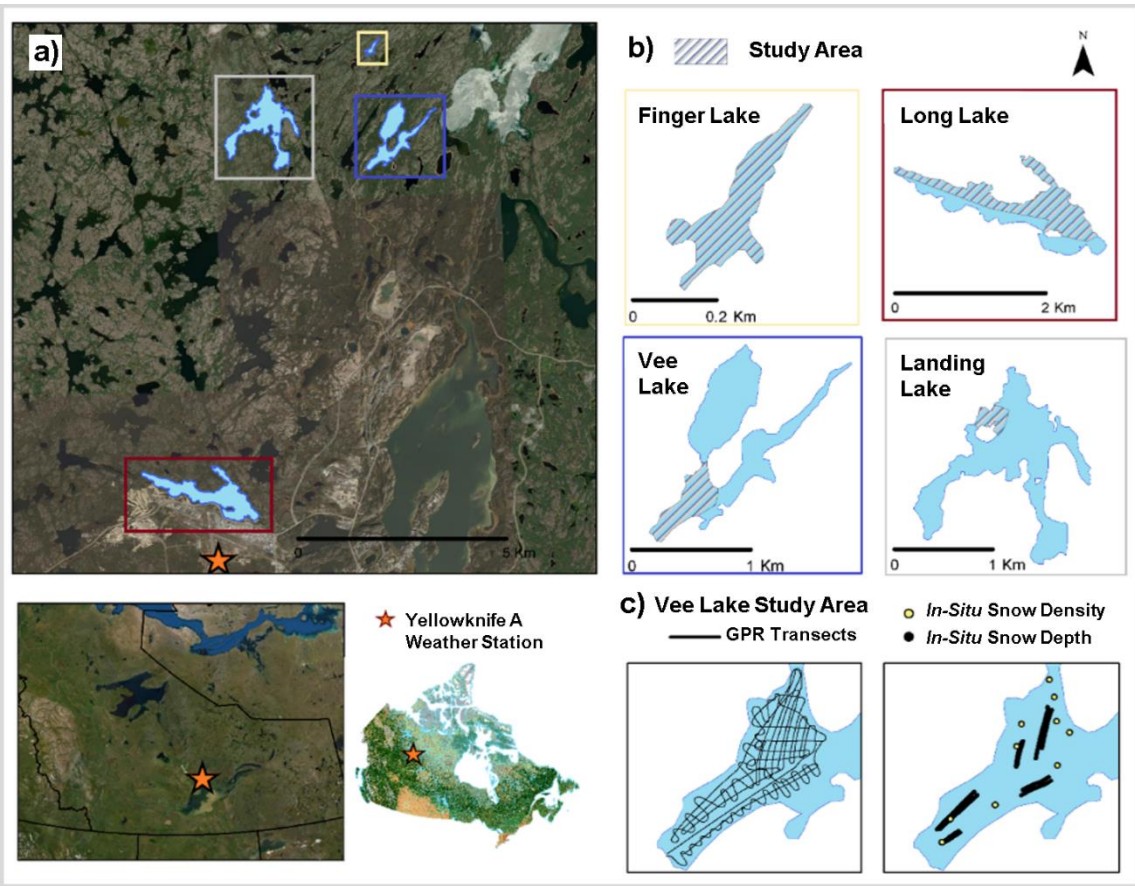


**Figure 1: This study focuses on (a) four lakes located north of Yellowknife, NWT, Canada, (b) Landing Lake, Finger Lake, Vee Lake, and Long Lake, shown on different scales depicting the area data collection took place (shaded colour). (c) The location of the GPR transects (Left) and in situ snow depth and density measurements (Right) on Vee Lake. (Background imagery: ESRI 2022, Landcover source: CCRS and NRCan, 2020)**

**Table 1: Data collection occurred on four lakes during early winter (December 2021) and late winter (March 2022, Landing Lake only) season. The surface area ($A_o$), Shoreline Length ($S_L$) and $A_o/S_L$ ratio are reported based on the entire shape of the lakes and not the area surveyed.**

| Site | Date Visited | Latitude | Longitude | $A_o$ ($km^2$) | $S_L$ ($km$) | $A_o/S_L$ ($km^2/km$) |
|------|-------------|----------|-----------|----------------|--------------|------------------------|
| Finger Lake | 12/09/2021 | 62.5750 | -114.3587 | 0.04 | 1.44 | 0.03 |
| Long Lake | 12/12/2021 | 62.4772 | -114.4422 | 1.13 | 10.35 | 0.11 |
| Vee Lake | 12/14/2021 | 62.5555 | -114.3502 | 0.70 | 8.63 | 0.08 |

| Landing-D Lake | 12/07/2021 | 62.5587 | -114.4103 | 1.08 | 11.71 | 0.09 |
| Landing-M Lake | 03/27/2022 | | | | | |

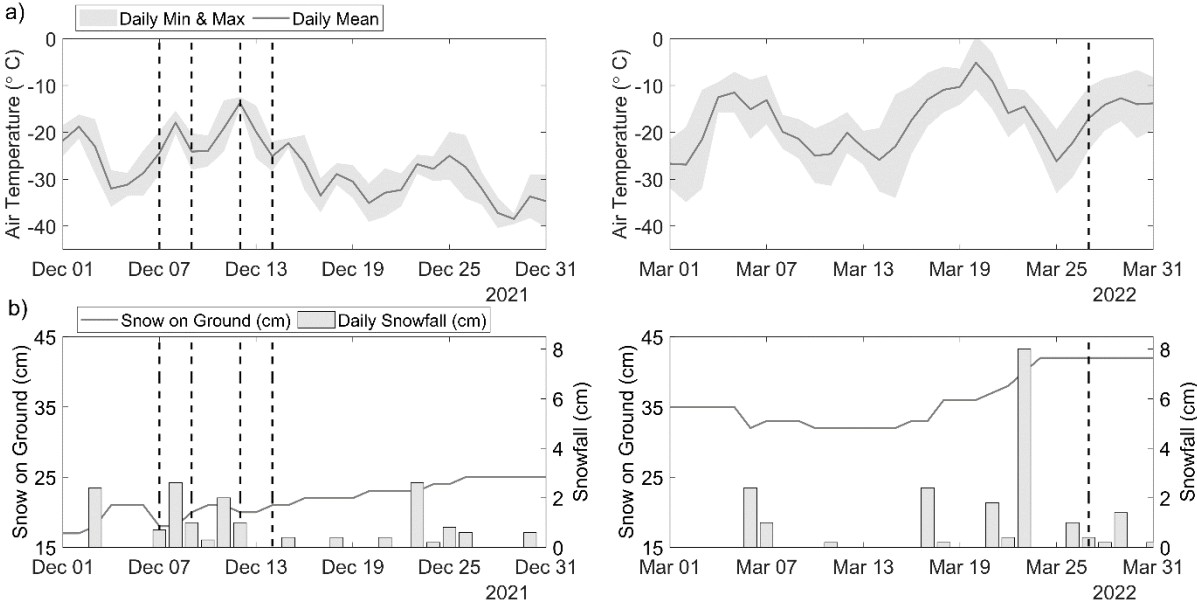

**Figure 2: (a) The daily mean, minimum, and maximum air temperatures, and (b) snowfall (bar) and snow on the ground (line)**
**collected at the Yellowknife weather station are shown for each day spent in the field (dash line).**

## 3 Methodology

### 3.1 GPR data acquisitions

GPR transects were acquired using the IceMap system (Sensors and Software Inc, 2022) paired with the 1000MHz Noggin sensor, with both the transmitting and receiving antennas oriented parallel at a fixed separation of 7.5cm. The IceMap system is configured with a GPS capable of recording location data simultaneously with the radar pulses, providing an accuracy of $\pm < 2$ m for the horizontal position. During the data acquisition, the IceMap GPR was set up in a sled pulled by a snowmachine. In the sled (Figure 3), the 1000MHz Noggin sensor was positioned behind the IceMap box and lined up with a Leica Global Navigation Satellite System (GNSS) Real-Time Kinematic (RTK) rover (Leica Geosystems, 2018). Using the GNSS RTK rover, the location data was recorded at a higher accuracy, which was later processed and paired with the GPR pulse locations. This process improved the coordinate accuracy in 3-dimensions to $\pm < 0.02$ m (see Sect. 3.3.2). While traveling at ~ 4 m/s, the resulting GPR trace spacing were ~ 9 cm, dependent on any slight changes in the speed of the snowmachine. The average footprint of each collected trace on all four lakes in December was 19 cm, and 30 cm in March on Landing Lake based on the

diameter of the first Fresnel zone (Fediuk et al., 2022). In considering the ~9 cm trace spacing to the footprint of each trace, the data results in over 50% overlap. The vertical imaging resolution was estimated at 6.5 cm on average across all four lakes
based on the one-quarter wavelength Rayleigh criteria using the 1000 MHz sensor (Kallweit and Wood, 1982), which has a vertical sampling interval of 0.1 ns. Approximately 38 km of GPR data was acquired over the four lakes initially traversed between December 7th to 14th, 2021 and an additional 6 km in March 2022, when revisiting Landing Lake. The transects were created following a gridded pattern to best cover the study area.

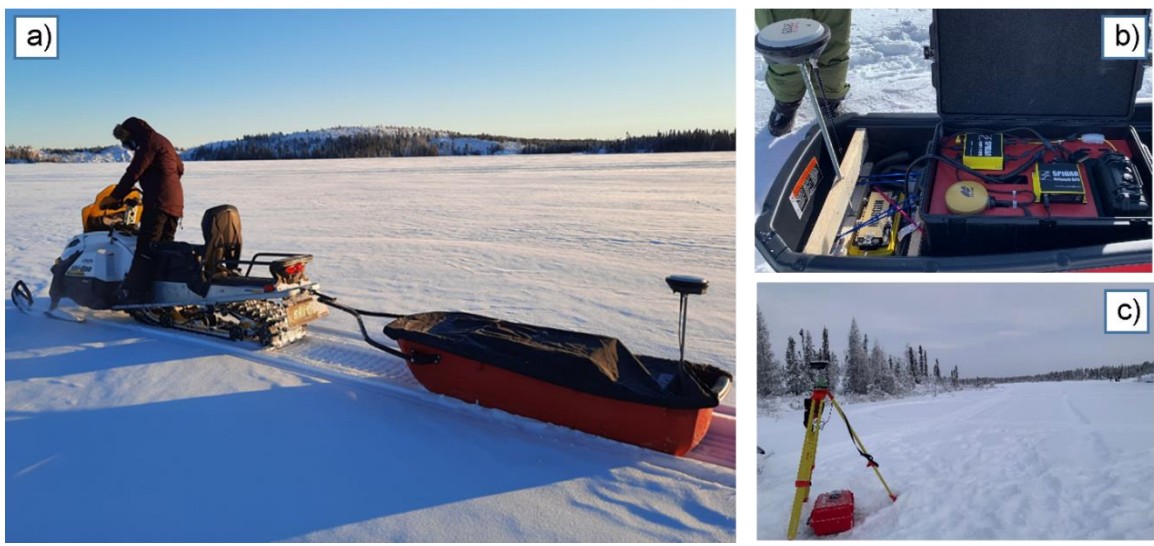

**Figure 3: (a) The GPR was pulled by a snowmachine. (b) The 1000 MHz sensor was paired with the GPR and an external GNSS rover recorded data simultaneously, to improve the spatial accuracy of the collected transects. (c) A local base station was set up on the lake for GNSS post-processing.**

**3.2 In situ observations**

In situ snow depth and density observations were gathered across areas of undisturbed snow and close to the GPR transects,
as shown in Figure 1c. Snow depths (Table 2) were collected using a SnowHydro Magnaprobe (Sturm et al., 1999; SnowHydro, 2013) along grids or transects across the lake, with the average spacing varying between lakes (~ 2.5 m). The spatial accuracy for the magnaprobe GPS receiver with use in the Arctic has been reported as ± 5 to 10 m (Walker et al., 2020), with a 0.01 m depth precision (Sturm and Holmgren, 2018). With known limitations in the Magnaprobe GPS accuracy, we used the RTK GNSS rover to measure the location of 291 magnaprobe measurements spaced out along the sampling transects on three of the
four lakes (Landing, Finger, Vee). We found the error from the magnaprobe GPS to be between 1.72 m to 8.43 m, with a mean (± standard deviation) error of 4.44 ± 1 m. On Landing-D Lake, there was frequent snow depression of the magnaprobe basket (~2 cm on average), where it sat below the snow surface. To account for the depression of the magnaprobe basket, we have corrected the in situ data for Landing-D Lake by 2 cm.

For each lake, the bulk snow density was sampled at 6 to 10 locations through measuring the specific snow volume and weight
of the vertical snow profile using a 5 cm diameter snow tube and an electronic scale with a 1g accuracy. The bulk snow
densities measured on each lake were averaged (Table 2) and used as a guide in determining the appropriate density to use for
deriving the snow depth. With limitations in fully capturing the variability of density across each area of focus, in later steps
(see Sect. 3.3.4), we applied densities within ±1 standard deviation of the mean to derive the snow depth from the GPR TWT.

**Table 2: In situ snow depth, $h_s$ and density, $\rho$ measurements were taken on the four lakes in December 2021 and, March 2022 on**
**Landing-M Lake. The density and snow depth varied between the four lakes (r = range, σ = standard deviation, n = count).**

| Site | | | Mean | Min | Max | r | σ | n |
|---|---|---|---|---|---|---|---|---|
| Finger Lake | $\rho$ | kg/m$^3$ | 160 | 140 | 190 | 50 | 15 | 10 |
| | $h_s$ | cm | 13.52 | 4.84 | 18.48 | 13.54 | 2.73 | 583 |
| Long Lake | $\rho$ | kg/m$^3$ | 245 | 180 | 310 | 130 | 47 | 7 |
| | $h_s$ | cm | 13.98 | 6.12 | 23.78 | 17.66 | 3.29 | 475 |
| Vee Lake | $\rho$ | kg/m$^3$ | 195 | 160 | 270 | 90 | 34 | 8 |
| | $h_s$ | cm | 16.09 | 6.29 | 21.00 | 14.71 | 2.48 | 427 |
| Landing-D Lake | $\rho$ | kg/m$^3$ | 170 | 140 | 200 | 60 | 21 | 6 |
| | $h_s$ | cm | 10.21 | 4.34 | 18.89 | 14.55 | 2.33 | 617 |
| Landing-M Lake | $\rho$ | kg/m$^3$ | 220 | 182 | 300 | 118 | 36 | 10 |
| | $h_s$ | cm | 35.61 | 24.70 | 50.81 | 26.02 | 4.54 | 595 |

## 3.3 Snow depth retrievals for GPR data

### 3.3.1 GPR signal processing

The snow-ice interface is challenging to identify due to interference between the direct wave and the reflection from the
shallow snow-ice interface, in addition to the noise caused by wavefield scattering and antenna bounce. To account for this,
signal processing was applied to the radargrams to remove any noise before automatically picking the TWTs. Initial processing
consisted of applying a de-WOW filter (band-pass filter with a mean subtraction) to the measured amplitudes for each trace
(Gerlitz et al., 1993). Next, a time-zero correction was applied to correct the first break times to ensure the snow surface was
set to zero nanoseconds (Ihamouten et al., 2010). Followed by a background median subtraction filter, which removed the
coherent "ringing" noise and the direct arrivals that masked the shallow reflections (Kim et al., 2007). Additionally, trace
stacking was applied to smooth the image (Yilmaz, 2001). All post-processing of the radargrams was conducted in MATLAB.

### 3.3.2 GPR trace location correction

Through simultaneously collecting spatial data using the RTK rover during the GPR data acquisition, the timestamps from
both the RTK GNSS and GPR GPS were used to pair the points and replace the spatial data of the GPR with the location

recorded from the RTK GNSS. The RTK GNSS spatial data (X, Y, Z) was set to collect every 0.5 m for each lake. To account

for the lower collection frequency of the RTK rover, the GPR traces that were not paired with an RTK GNSS point were linearly interpolated. In comparing the accuracy of the GPR GPS to the RTK GNSS for the paired locations, the error in GPS accuracy (easting & northing) was between 0.22 m to 4.97 m, with a mean Euclidean difference of 2.63 ± 1.21 m.

### 3.3.3 Automatically picking GPR TWT

The GPR TWTs were extracted using the modified energy ratio algorithm (Wong et al., 2009), which automatically picks the

first break. With the input of an estimated depth and wave speed, the picker is guided to a region of the time window and picks the first initial zero crossing of the wavelet reflection, identifying the TWT. The radargram after signal processing can be seen in Figure 4a showing the TWT automatic picks along a transect on Landing-D Lake. Viewing Figure 4b as a function of elevation (meters above sea level), the variation in snow surface and thickness as well as ice surface can be seen.

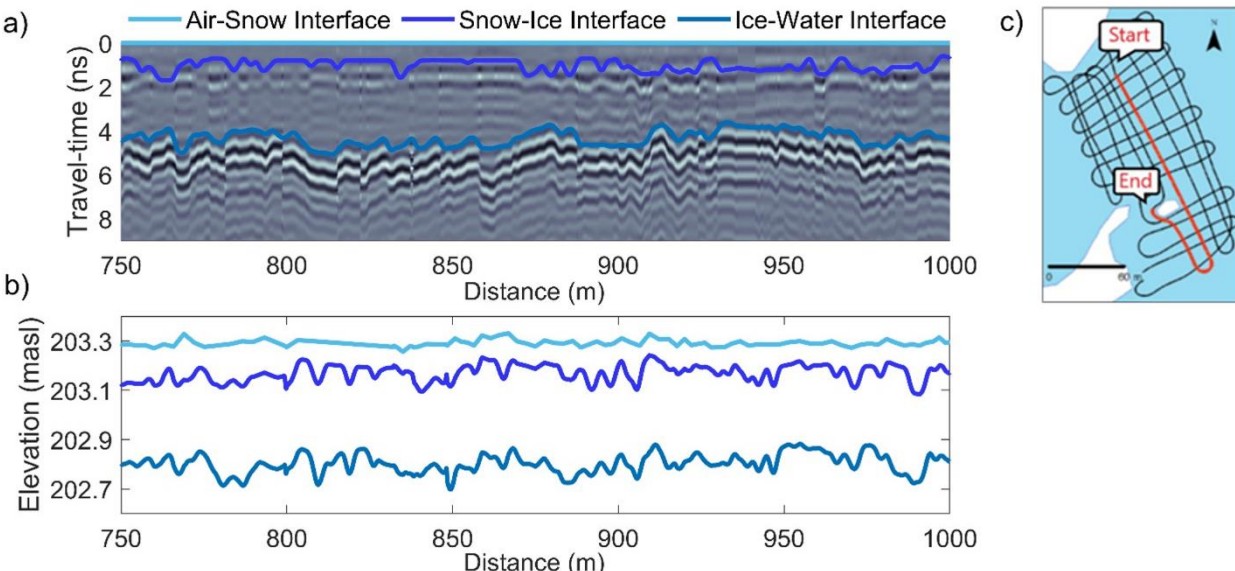

**Figure 4:(a) After applying signal processing, the modified energy ratio algorithm was used to automatically pick the TWTs. The air-snow interface is represented at time-zero and the snow-ice and ice-water interfaces were picked using the first initial zero crossing of the wavelet reflection. (b) The automatic TWT picks are shown as a function of elevation, where the variability in snow surface, ice surface, and the ice bottom can be seen. (c) The location for this 250 m example is on Landing-D Lake.**

### 3.3.4 Calculating snow depth from TWTs and density

Snow depth was derived using the automatically picked TWTs and the wave speed of the radar signal. To determine the wave speed of the radar signal traveling through the snow, the relative permittivity was calculated. There are several empirical

equations available for deriving the relative permittivity from snow density. Previous work (i.e., Di Paolo et al., 2018; Webb et al., 2021) found there is significant variability in deriving permittivity between these equations for larger snow densities, however, based on the snow densities presented within this study, there is less variability between equations. Therefore, the Kovacs et al. (1995) equation is used to calculate the relative permittivity. The measured in situ snow density within a range of one standard deviation of the average for each lake (Finger = 175 kg/m$^3$, Long = 245 kg/m$^3$, Vee = 195 kg/m$^3$, Landing-D = 190 kg/m$^3$, Landing -M = 200 kg/m$^3$) was used to calculate relative permittivity using Eq. (1) as:

$$\varepsilon_r = (1 + 0.845\rho)^2 \tag{1}$$

where $\rho$ is the density of snow, $\varepsilon_r$ is the relative permittivity. As the wave speed ($V$) at which the EM wave moves through snow depends on the snow relative permittivity, $V$ was calculated using Eq. (2) as:

$$V = \frac{C}{\sqrt{\varepsilon_r}} \tag{2}$$

where $C$ is the speed of light (0.3 m/ns) and $\varepsilon_r$ is the relative permittivity. The wave speed, $V$ is calculated for each lake (Finger = 0.261 m/ns, Long = 0.249 m/ns, Vee =0.258 m/ns, Landing-D = 0.259 m/ns, Landing-M =0.257 m/ns) and therefore, snow depth ($h_s$) was derived using Eq. (3) as:

$$h_s = \frac{V \times TWT}{2} \tag{3}$$

where TWT is the two-way GPR travel-time.

### 3.4 Comparing GPR TWT derived snow depth to in situ snow depth

Derived snow depths from GPR TWTs were compared to in situ snow depth measurements collected during fieldwork. Around each measured in situ snow depth, the GPR traces that fell within a 6 m radius were used to compare the accuracy of the derived snow depth. The 6 m radius was chosen due to the location accuracy calculated with the in situ snow depth observations (mean error of 4.44 ±1 m; see Sect. 3.2). The snow depths were derived in two different scenarios: 1) *closest match*, where the single closest matched snow depths within the 6 m radius was selected, and 2) *distance weighting,* where the closest 50 % of total matched snow depths within the 6 m radius were selected and distance weighted. The removal of 50 % minimizes the selection of GPR traces over the 6 m span and accounts for the spatial variability in snow depth expected over this 6 m length scale.

# 4 Results

## 4.1 Snow depth from GPR TWT

Collected GPR data across the four lakes traversed in December 2021 resulted in 406,164 derived snow depth observations (Figure 5). The GPR-derived snow depths ranged from ~ 7 cm to 25 cm (Table 3), with the shallowest mean snow depth observed on Landing-D Lake on December 7th at $12.76 \pm 3.25$ cm, and the deepest mean snow depth on Vee Lake on December 14th at $16.06 \pm 3.08$ cm. The GPR transects on Landing-D Lake covered the smallest area of focus relative to the size of the lake (2.5%) and distance traversed (~ 3 km) and showed snow depth variability of 15 cm around islands, open areas, and shorelines. The entirety of Finger Lake ($A_o = 0.04$ km$^2$) was traversed on December 9th, where the deepest snow depths were observed along shorelines (max = 24.83 cm), compared to the open stretch of the lake (min = 6.53 cm). Collected snow depth data on Long Lake on December 12th showed the largest spatial area, spanning 3 km from northwest to southeast, with a total distance covered of 16 km. Long Lake showed the largest range in snow depth (6.21 cm to 22.34 cm) and density (180 kg/m3 to 310 kg/m3).

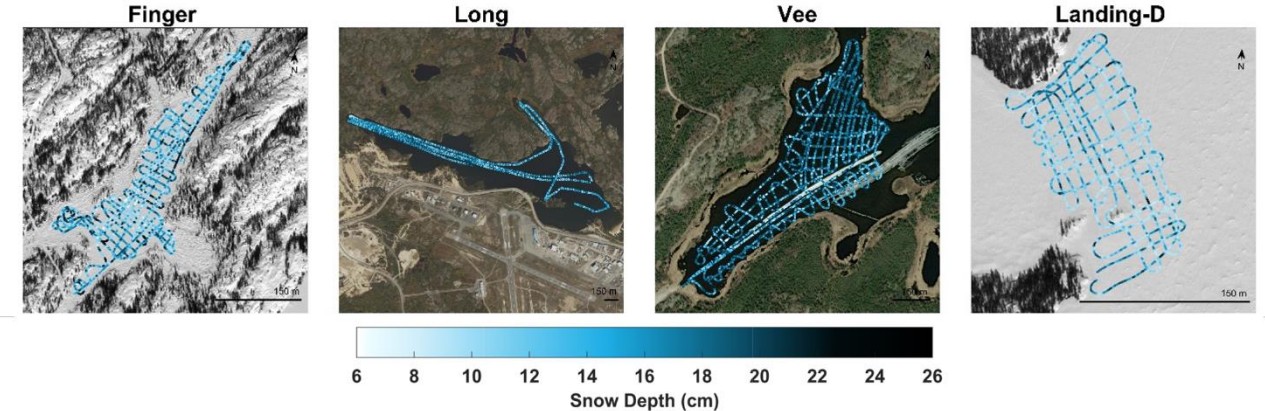

**Figure 5: Maps show the 406,164 GPR-derived snow depth observations along the transects over each lake for December 2021. (Background imagery: ESRI 2022)**

**Table 3: The GPR TWT-derived snow depth statistics from the four lakes during December 7– 14, 2021 (r = range, σ = standard deviation, n = count, d = distance traversed, and s = average trace spacing)**

| Site | Mean (*cm*) | Min (*cm*) | Max (*cm*) | r (*cm*) | σ (*cm*) | n | d (*km*) | s (*m*) |
|---|---|---|---|---|---|---|---|---|
| Finger Lake | 14.60 | 6.53 | 24.83 | 18.29 | 3.55 | 63589 | 5.36 | 0.08 |
| Long Lake | 14.68 | 6.21 | 22.34 | 16.13 | 3.29 | 152554 | 16.27 | 0.11 |
| Vee Lake | 16.06 | 6.44 | 23.18 | 16.74 | 3.08 | 151853 | 12.72 | 0.08 |
| Landing-D Lake | 12.76 | 7.60 | 22.42 | 14.67 | 3.25 | 38168 | 3.06 | 0.08 |

## 4.2 Comparing GPR vs. magnaprobe snow depths

The in situ snow depth observations (n =1932) were used for all four lakes to validate the GPR-derived snow depth in December
2021. The comparison of in situ and GPR-derived snow depths for scenarios 1 and 2 are shown in Figures 6 and 7. We found
that the minimum error snow depth exists within a 6 m radius ($R^2$ = 0.9, RMSE = 0.7 cm, MAE = 0.3 cm on average) for all
four lakes (Figure 6a). The distance of each minimum error pair was, on average, 3.79 ± 1.5 m apart, compared to the measured
accuracy error with the magnaprobe (4.44 ± 1 m). Through identifying the distance between each GPR and in situ snow depth
pair, we confirmed that the GPR measurements further away (within 6 m radius) from the in situ snow depth are the appropriate
pairs in most cases (Figure 6b). Therefore, we applied scenario 2 to evaluate the accuracy of the GPR-derived snow depths
(Table 4) and applied scenario 2 for further data analysis.

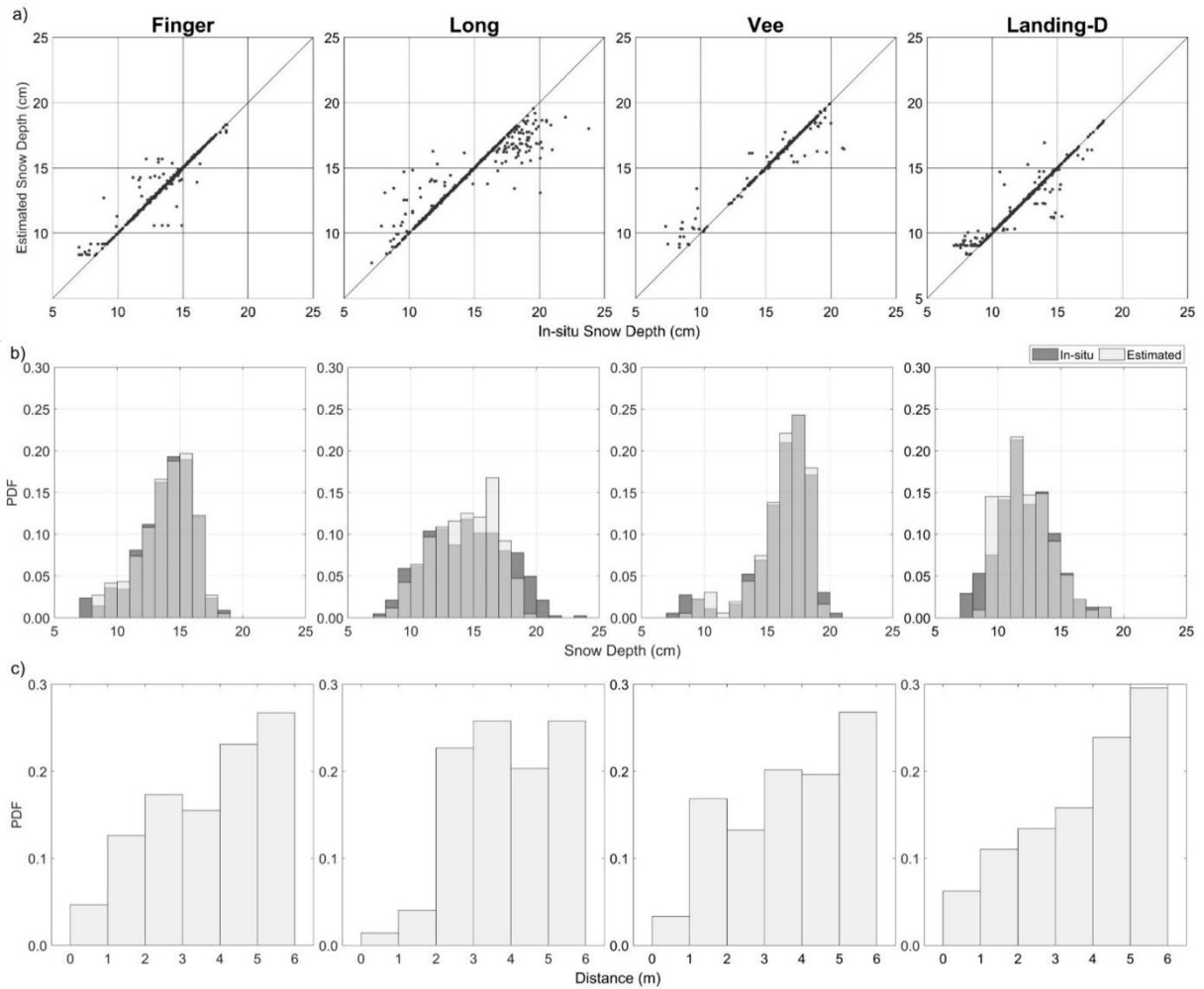

**Figure 6: Validation of GPR-derived snow depths using a 6m radius, Scenario 1 *(closest match),* (a) scatterplots and (b) histograms of the in situ and GPR-derived snow depth, and (c) bar plots of the distance from the paired in situ to GPR-derived snow depth.**

Scenario 2 showed strong agreement between the in situ and estimated observations (Figure 7) with $R^2 = 0.6$, RMSE = 1.6 and MAE = 1.0 cm on average for all lakes. If considering all GPR-derived snow depth observations within the 6 m radius there is minimal differences in the validation statistics (RMSE = 2.7 cm, MAE = 2.0 cm, Bias = 0.13 cm), however Scenario 2 is used for further analysis due to the variability in snow depth seen within the 6 m radius (2.1 cm to 4.9 cm). In using scenario 2, Long Lake showed the lowest agreement ($R^2 = 0.5$, RMSE = 2.2 cm, MAE = 1.5 cm) with the GPR-derived snow depth showing slight over and under estimations. The strongest agreement was found on Vee Lake with $R^2 = 0.7$, RMSE = 1.4 cm,
and MAE = 0.8 cm. The relative error of the GPR-derived snow depth was 8 % on average for all four lakes traversed in December, with Vee Lake being the most accurate (relative error = 6 %) and Long Lake the least (relative error = 11 %). The snow depth derived from the GPR-TWT was consistently overestimated when compared to in situ observation for shallower snowpacks (<10 cm) across all four lakes.

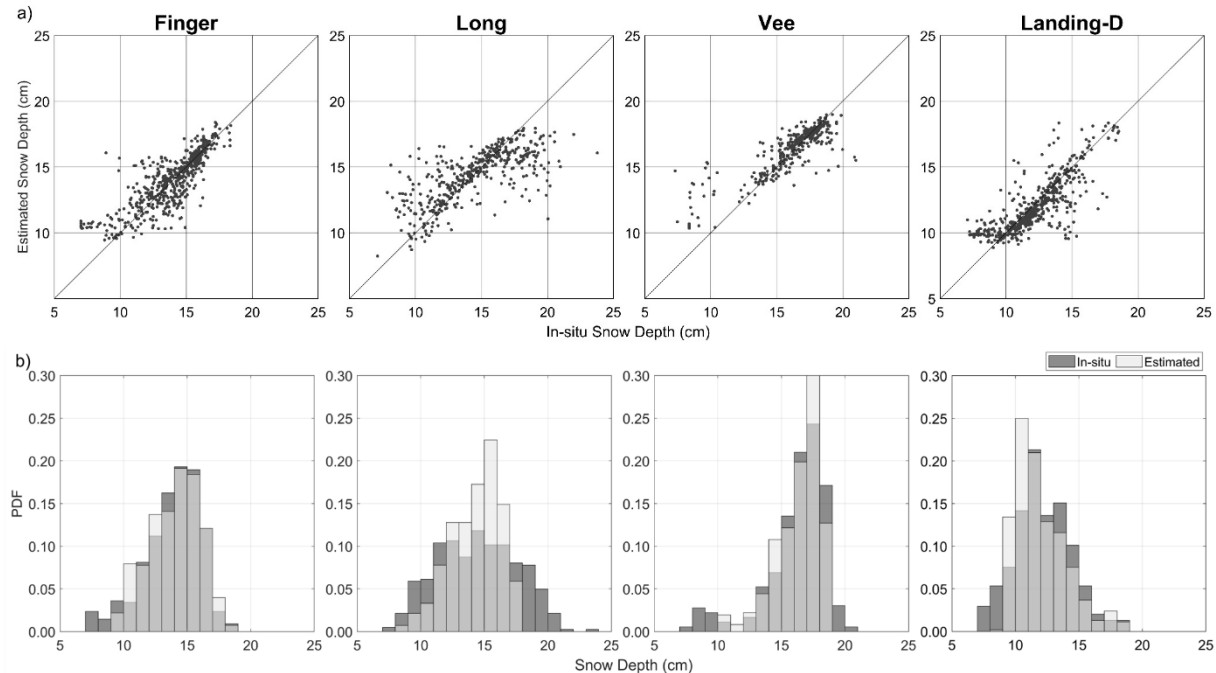


**Figure 7: Validation of GPR-derived snow depths using a 6m radius, Scenario 2 (*Distance weighting*), (a) scatterplots, and (b) histogram of the in situ and GPR-derived snow depth.**

**Table 4: Statistics of GPR derived snow depths versus the magnaprobe collected snow depths for Scenario 2.**

| Site | $R^2$ | MAE (*cm*) | RMSE (*cm*) | Bias (*cm*) | Relative Error (%) | n |
|------|-------|------------|-------------|-------------|---------------------|---|
| Finger | 0.7 | 0.9 | 1.3 | 0.1 | 8 | 554 |
| Long | 0.5 | 1.5 | 2.2 | -0.1 | 11 | 472 |
| Vee | 0.7 | 0.8 | 1.4 | 0.2 | 6 | 362 |
| Landing-D | 0.6 | 0.9 | 1.4 | -0.2 | 8 | 544 |
| **Mean** | **0.6** | **1.0** | **1.6** | **0.01** | **8** | |

## 4.3 Snow depth mapping

Snow depth distribution maps were generated at a 1 m resolution through interpolating (inverse-distance weighting) the GPR-derived snow depth observations (Figure 8). Through re-gridding to 1 m resolution and interpolating, the snow depths ranged from 8 cm to 22 cm in December 2021. The deepest snowpack, on average, was observed on Vee Lake (15.99 ± 0.79 cm), ~ 4 cm deeper than Landing-D Lake (12.73 ± 0.87 cm) during December 2021 field campaign. The interpolated GPR-snow depths consistently show an increase in snow depth variability closer to the lake perimeter compared to areas farther from the

shoreline and closer to the center of the lake. The snow depth on Finger Lake showed a decrease of ~2 cm per meter as the distance from the perimeter increased, however, this was not observed on the additional lakes. Transect profiles (Figure 8) created over the 1 m resolution snow depth maps show an example of the variability in snow depth across each lake. The spatial correlations of the 1 m resolution snow depths from the GPR transects were estimated using an experimental semi-variogram that was fit using an exponential model (Figure 9). The largest correlation length was observed on Vee Lake (11.25

m) in December 2021, and Landing-M Lake (18.18 m) overall. The correlation length on Landing-D Lake in the early season was measured at ~10 m less than that of the late winter season, while Long Lake showed the smallest distance, at 6.42m over the largest spatial area.

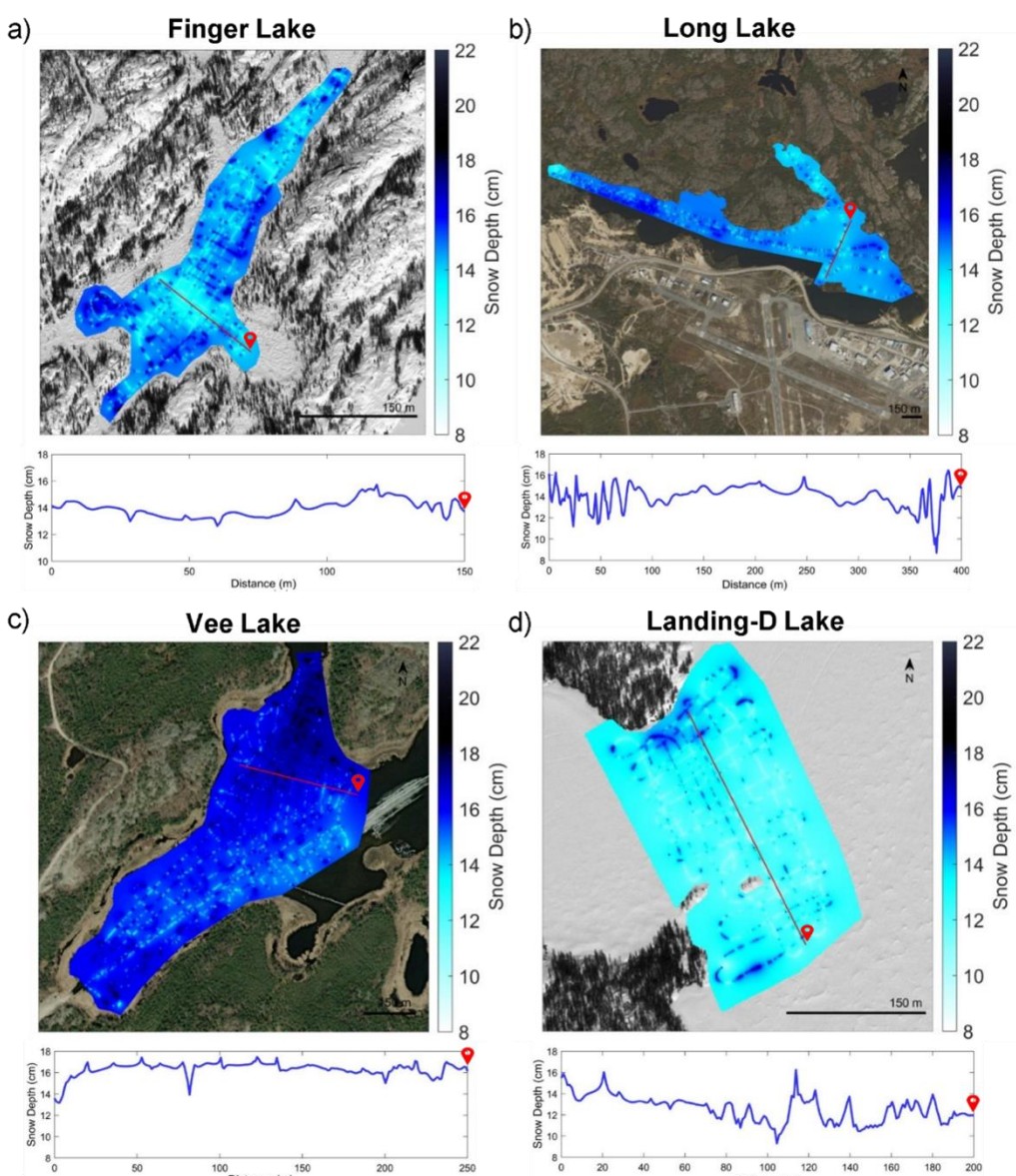

**Figure 8: Maps show the GPR-derived snow depth using an inverse-distance weighted model to interpolate the snow depth over (a) Finger Lake, (b) Long Lake, (c) Vee Lake, (d) Landing-D Lake at 1 m resolution and showing a transect profile across a portion of the lake (profile transect ends at the red symbol marked on each lake, Background imagery: ESRI 2022).**

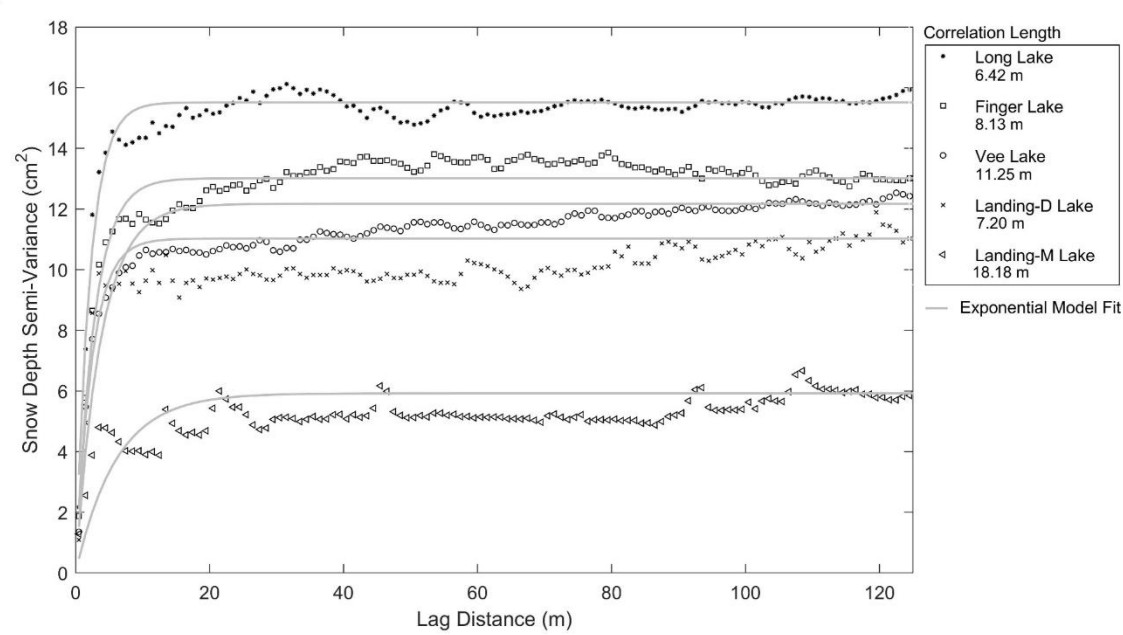

**Figure 9: The experimental variograms for GPR-derived snow depth transects were fit to an exponential model to determine the correlation length.**

## 4.4 Early vs. late winter season

Landing Lake was revisited for the data collection on March 27th, 2022, resulting in an additional 73,732 snow depth observations from GPR TWTs over ~6 km (Figure 10). In December 2021 and March 2022, the snow depth derived on Landing Lake varied, with Landing-D Lake ranging from ~8 to 22.50 cm and Landing-M Lake from ~10 to 50 cm. The snow depth was, on average, 12.76 (± 3.25) cm in December and more than twice that in March (35.83 ± 2.54 cm). The snow density in the early season was, on average, 170 kg/m$^3$, whereas in the later season measured at an average of 220 kg/m$^3$ (Table 2). The results showed that agreement between in situ snow depth observations and Landing-M Lake GPR-derived snow depth ($R^2$ = 0.66, RMSE = 2.9 cm, Bias = 0.4 cm, n = 498) was not significantly improved when compared with Landing-D Lake (Figure 10). However, the relative error was improved on Landing-M Lake with a deeper snowpack (5%) than that of Landing-D Lake (8%). The GPR could derive the minimum snow depths seen on Landing Lake during the later season, as opposed to that in the early season, where the GPR-derived snow depth could not capture the shallowest snow area (4.5 – 10 cm).

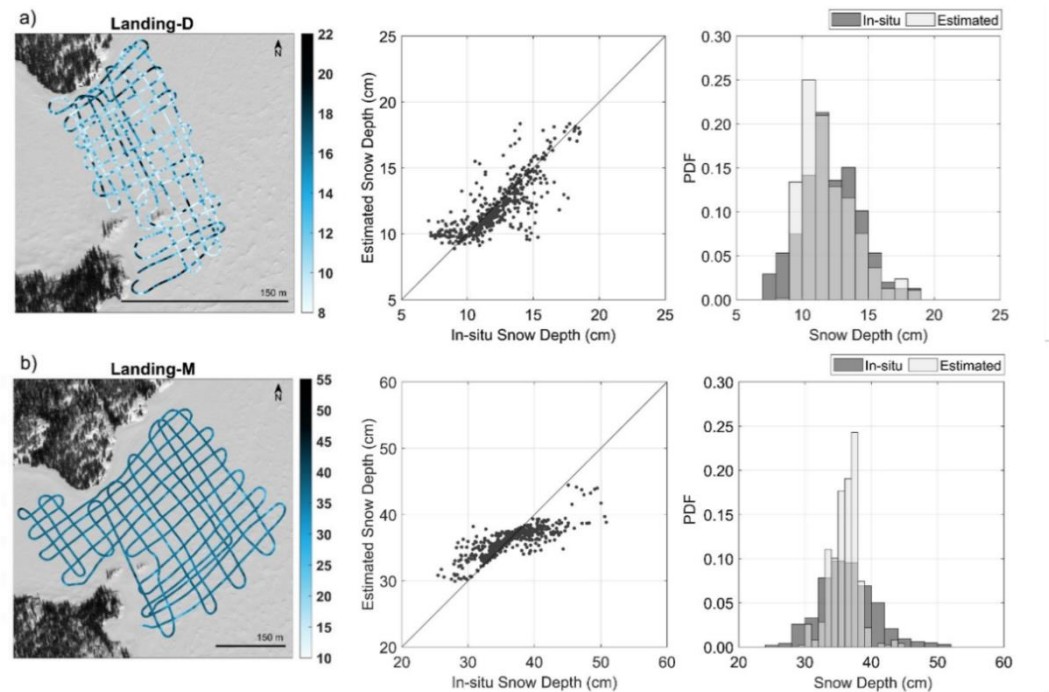

**Figure 10: Maps show the GPR derived snow depth on GPR transects, and the scatterplot and bar plot compare in-situ data vs GPR-derived snow on (a) Landing-D Lake during December 2021 and (b) Landing-M Lake during March 2022 (Background imagery: ESRI 2022)**

In comparing the difference in snow depth and snow density over the winter season, Figure 11 shows IDW 1-m snow depth maps and snow density maps (created using the in situ observations). The snow density from early season to late winter season increased between 10 to 80 kg/m³, while the snow depth increased in areas by 18 to 28 cm. There were no surveyed areas on the lake that experienced a decrease in snow density or depth based on the two field sampling dates. Areas with a shallower snowpack in December 2021 saw the largest increase in snow depth by March 2022 ($R^2 = 0.57$), which agrees with the decrease

in snow depth variability noted in Figure 9 by the correlation lengths. Additionally, the largest increase in density from early to late winter season occurred closest to the shoreline. More densification occurred on areas that were less dense than areas that had a higher density in December 2021 by March 2022 ($R^2 = 0.59$). In exploring the change in snowpack over the winter season, we found no spatial relationship between changes in the depth and density across the area surveyed on Landing Lake.

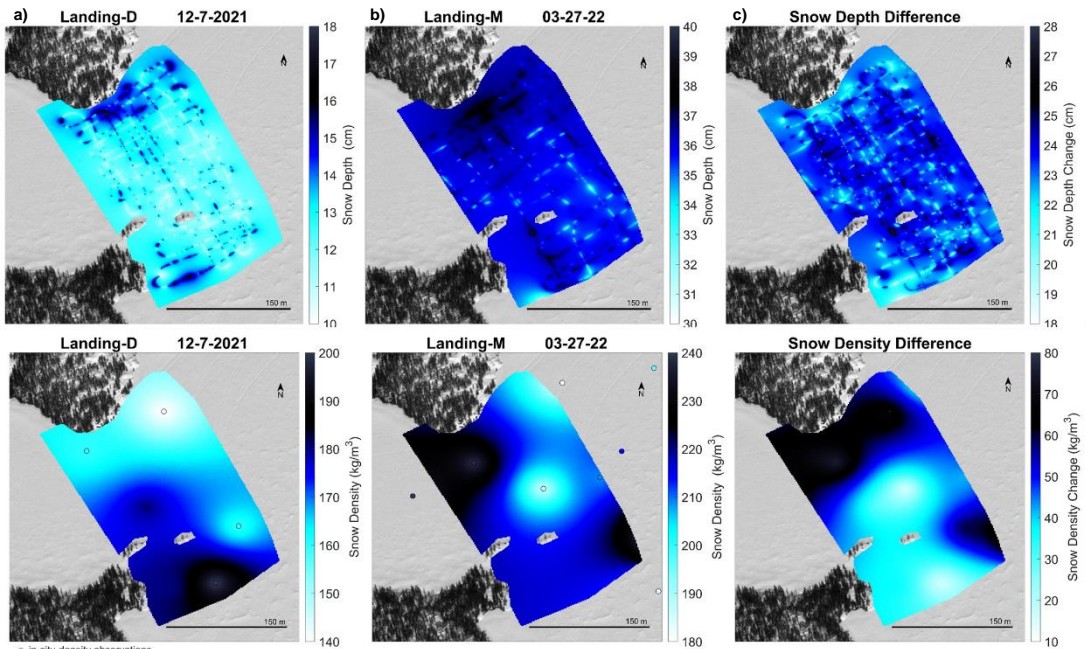

**Figure 11: Maps of Landing Lake snow depth (top) and density (bottom) in (a) December, (b) March and (c) the difference between the two were created using IDWs of the GPR-derived snow depth and the in situ snow density observations.**

## 5 Discussion

This study demonstrates the effectiveness of using GPR to derive snow depth over lake ice, where the snowpack is generally not deep, and it is challenging to capture the snow-ice interface accurately. The findings of this study enhance our ability to
collect accurate and efficient snow depth observations over large areas of lakes, which are comparable with previous studies using GPR TWT over land and sea ice (Pfaffhuber et al., 2017; McGrath et al., 2019). This automated method successfully derives snow depth over lake ice from GPR, making it a valuable tool for estimating and analysing the thermal energy balance of the ice surface across the entire lake and gaining a better understanding of the physical processes involved in snow distribution.
Freeze-up for lakes surrounding Yellowknife typically occurs during October. However, during the 2021 to 2022 season, freeze up was reported to occur later compared to the 2018 to 2020 seasons based on Yellowknife's snowmobile association data. October air temperatures reported at the Yellowknife weather station showed a mean temperature increase of 4.4°C between 2020 (-1.85°C) and 2021 (2.6°C), with a 3.18°C increase compared to the 5-year and 10-year October mean air temperatures. During the 2021 to 2022 water year, around 75 cm of snowfall was reported by the Yellowknife weather station,
which accounted for 46% of the total annual precipitation. The 2021 to 2022 water year had 20% less snowfall compared to the previous water year of 2020 to 2021, which experienced around 93 cm of snowfall (76% of total precipitation). On average,

in the past 5 to 10 years, 40 to 45% more snowfall was reported than in the 2021 to 2022 year. The timing and amount of snowfall influences the composition, thickness, and phenology of lake ice. Increased snow accumulation in early season and on thin lake ice that has reduced buoyancy will create leads and cause overflow (the upwelling of water from the water column), which increases the likelihood of snow ice growth. Thin and patchy snow ice (0–4 cm) was observed on the lake ice surface during the December and March field campaigns, comprising 0% to 6% of the lake ice composition. Based on observations up until March 2022, minimal amounts of snow ice were found, indicating that there was little overflow that occurred on these four lakes during the winter season prior to the beginning of ice break-up. In December 2021 and March 2022, the lakes consistently showed a shallow snowpack on average (Table 3) than snow on the ground over land (Figure 2) reported at the nearby Yellowknife weather station. The lakes had an average of 24% to 29% less snow than measured over land in December 2021, and 15% less in March 2022. Thus, using snow depths measurements from land as input to lake ice models will overestimate lake snow depth by a seasonally dependent factor and affect the modeled ice thickness (Kheyrollah Pour et al., 2017).

Snow dunes are formed from snow redistributed by wind on relatively level ice surfaces and in turbulent wind fields. The accumulation of snow over the lakes varied during the study period, which could be explained by the total snowfall (8 cm) with consideration to wind redistribution and compaction seen between December 7$^{th}$ (Landing-D Lake $\overline{h}_s$ = 12.76 cm) to December 14$^{th}$ (Vee Lake $\overline{h}_s$ = 16.06 cm). Snow dunes were present across all the lakes during both field campaigns. This study explored the distribution of snow over each lake (Figure 9), which revealed local-scale variability of snow depths from redistribution of the snow across all the lakes (correlation lengths between 6–19 m). The semi-variogram analyses applied determined the horizontal spacing of the snow dunes and found Long Lake to have the shortest correlation length (6.42 m). Landing Lake was observed to have an increase in correlation length throughout the winter season from ~7 m to ~19 m. These inferred variability length-scales are similarly supported in the literature, reporting correlation lengths from 5 to 20 m (Gunn et al., 2021a; Sturm and Liston, 2003).

In comparing the spatial snow depth variability across the four lakes, the physical characteristics of Long Lake were found to explain the reduced correlation length in comparison to the three additional lakes. Long Lake has the largest shoreline length to surface area and spans ~3 km northwest to southeast, resulting in the largest wind fetch area and the higher snow density compared to the additional three study areas. While on Landing Lake, both the snow depth and density increased over the season, however, more frequent sampling dates would be necessary to determine the reason for the decrease in snow depth variability from December to March between early and late season.

The lower accuracy in GPR-derived snow depths on Long Lake (± 11%) could be attributed to the use of a radius to compare the derived and in situ snow depths that was approximately the same magnitude as the length scale of snow depth variability. Vee Lake had the highest accuracy (± 6%) in deriving the snow depth and the largest correlation length (~11 m) in December 2021. The greatest accuracy (± 5%) was found during the late season on Landing-M Lake, which was also found to have the largest correlation length (~19 m). Therefore, the snow depth variability within 6 m was less on Vee Lake and Landing Lake

than on Long Lake. The accuracy of this method may be improved by enhancing the spatial location of the in situ snow depth measurements and sampling more frequently within the length scales of each lake. In improving the spatial accuracy of the in situ snow depth observations and the frequency of measurements, the accuracy of this method can be further assessed.

The snow distribution over lake ice is known to be affected by wind and surrounding vegetation (Adams, 1976a). In this study, weak relationships were found between the lake snow depth and distance to shoreline perimeter. On Finger Lake, where there was complete coverage of the lake, the snow depth declined ~2 cm per meter from the shoreline to the centre of the lake, but no change was observed on the other three lakes. This lack of data representativeness around the shoreline and the difficulty associated with maneuvering the snowmobile in the deep, lighter snow at slow speeds, or the turbulent winds affecting which shoreline the snow will be distributed along may be responsible for this discrepancy. Winds reported at the Yellowknife weather station reached speeds above the ~4 to 11 m/s threshold required to transport snow (Li and Pomeroy, 1997), however, with the majority of strong winds coming from the northeast and northwest, our lack of data on the southern perimeter on each lake may also affect our findings.

During the field campaigns, both the 1000 MHz and 500 MHz GPR antennas were used. Our findings indicated that the 1000 MHz sensor is more suited for estimating shallow snow depths more accurately, particularly during the early season (not shown) due to the shorter wavelength and the higher vertical imaging resolution associated with the 1000 MHz sensor. This study revealed a limitation in deriving snow depths below 7 cm using the 1000 MHz sensor, identifying a threshold. This finding is consistent with previous studies (Pfaffhuber et al., 2017), and as such, the in situ observations below 7 cm were excluded from the validation analysis. During the March 2022 campaign, seldomly snow depth was observed below 25 cm, meaning the vertical imaging resolution of 6.5 cm for the 1000 MHz sensor did not limit our data acquisition.

The analysis showed that no correction is required for compaction caused by the GPR sled. In considering the crossover locations (n = 533) on each of the lakes, we assessed the difference in TWT between the initial pass and the second pass and found that the average TWT difference was $0.02 \pm 0.31$ ns. Given the average velocity of 0.26 m/ns for the four lakes, and applying the one-quarter wavelength Rayleigh criterion, the uncertainty of the TWT picks is approximately three samples (~0.3 ns). Therefore, the average TWT difference at crossover locations is within our uncertainty estimates of the TWT picks. In further exploring the change in TWT from the initial pass to the second, 56% of the observations show the TWT for the second crossover to be larger than the initial. We found that shallower snow depths (or smaller TWTs) resulted in a decrease in travel time for the second pass, while deeper snow depths (or larger TWTs) showed an increase for the second pass for both early ($R^2 = 0.30$, $p < 0.05$) and late winter season ($R^2 = 0.46$, $p < 0.05$). However, these trends do not show dependency on the total snow depth accumulated throughout the winter season, as the average crossover differences of the data collections for early and late seasons (shallow and deep snow depths) are unbiased. Overall, although there is a change in density on the sled track ($\bar{\rho}$ sled = $340 \pm 20$ kg/m3) compared to the density of the fresh snow (Table 2), the effects of a decrease in depth and increase in density under compaction from the snowmachine are naturally compensated and were confirmed with the crossover location TWT differences. The snow depth was measured at 1.5 cm less on average by using the density of the sled track for

depth estimation rather than fresh snow density. Therefore, the effect on GPR derived snow depth is minimal because minimal snow mass was lost.

Lake snow is not well characterized in the various dielectric permittivity models used for wave speed estimation. In this study we found the snow depth retrieval is weakly dependent on the choice of empirical equation used to derive the snow depth from density. Within our analysis we used the Kovacs et al. (1995) equation to derive the permittivity. In addition, we also tested different empirical relationships to calculate the permittivity (i.e., Robin et al., 1969; Robin, 1975; Tiuri et al., 1984; Stein et al., 1997; Frolov and Macheret, 1999, Webb et al., 2021) and found very slight differences in the dielectric constant, if any at

all. The results ($\overline{\varepsilon_r}$ = 1.37) from the Kovacs et al. (1995) method are identical to using the Robin et al. (1969), Robin (1975), Tiuri et al. (1984), Frolov and Macheret (1999), and very similar to Stein et al. (1997) equation ($\overline{\varepsilon_r}$ = 1.34), with the largest difference using the Webb et al. (2021) equation ($\overline{\varepsilon_r}$ = 1.29). In exploring the permittivity for the snow densities presented within this study (175 kg/m$^3$ to 245 kg/m$^3$), the numerous empirical relationships result in very similar permittivity's for these lower densities and sub-millimetre differences in the snow depth accuracy statistics (not shown). Di Paolo et al. (2018) shows

in comparing 19 different empirical formulas to calculate permittivity, there is less variability for lower densities than there is for higher density snowpacks (i.e., 300 kg/m$^3$ to 550 kg/m$^3$). Lake snow has generally been reported to be shallower and less dense than snow types used to parameterize these models. However, based on the agreement among models and the limited representation for a model based on lake snow observations we have sided with the Kovacs et al. (1995) equation.

Additionally, our study found that when deriving snow depth from GPR TWT, the sensitivity in derived snow depth due to

snow density was minor for shallow snowpacks. As a result, the impact of spatial density variability on the retrieved snow depth was minimal. The uncertainty in snow density, based on the mean and ± 1 standard deviation measured in the field, propagates as 0.16 to 0.50 cm uncertainty in GPR-derived snow depth in December 2021 and 0.90 cm in March 2022. Although snow density varies spatially in three-dimensions (King et al., 2020), we did not account for this variability in our study. However, due to the shallow nature of snow on the lake ice, we found that this effect of spatial density variability on the snow

depth retrieval was minimal, thus permitting the use of a uniform spatial density in deriving shallow snow depth from GPR.

**6 Conclusion**

GPR has proven an effective tool for mapping various components of the cryosphere including snow over land, glacial firn, sea ice, and lake ice thickness. However, from our knowledge no studies had applied GPR to derive snow depth over lake ice where there are challenges associated with capturing the shallower snow thickness. The snow over lake ice has commonly

been ignored when deriving lake ice thickness. In this study, we collected 1000 MHz GPR acquisitions over four neighbouring sub-arctic lakes and by applying a fully automated post-processing method, we accurately derived ~500,000 snow depth retrievals covering ~38 km in December 2021 and ~6 km in March 2022. The lake snow depths derived from GPR TWT resulted in an average relative error under 10% when compared to in situ observations for the early and late winter season.

These results suggest that GPR acquisitions can be used to derive lake snow depth and can substitute manual snow depth observations, requiring only an observation of dry snow density, or snow depth and the radar travel-time for calibration. The spatial variability of snow density and the choice of empirical relative permittivity equation had little affect on the derivation of relatively shallow lake snow depth using GPR TWTs. Overall, this method can ease data collection to assist in validating snow distribution models or remote sensing products, as well as input for climate, thermodynamic, and hydrological modelling. The four small lakes, Landing Lake (62.5587 °N, 114.4103 °W), Finger Lake (62.5750 °N, 114.3587 °W), Long Lake (62.4772 °N, 114.4422 °W), and Vee Lake (62.5555°N, 114.3502 °W), have varying morphometry in terms of the surface area and shoreline length. Findings suggest lakes with a larger surface area to shoreline length ratio have higher spatial variability when compared during the same time period. Full spatial coverage across each lake during data acquisition can lead to a better understanding of the impacts of wind and shoreline vegetation have on the spatial variability. Simultaneously collecting ice thickness observations further improves our understanding of the spatial relation between snow depth and ice thickness. The findings of this research can lead to an improved understanding of snow and lake ice interactions, which is essential for northern communities' safety and wellbeing and the scientific modelling community.

**Code and data availability**

Data will be available through the Government of Northwest Territories' Discovery Portal (GNWT Discovery Portal) as well as ReSEC Lab data portal - **DOI (will be replaced after review process with public DOI)**. All code used for data processing and analysis of this study are available from the corresponding author upon request. This manuscript is a slightly modified version of AP master thesis (Pouw, 2023).

**Author contributions**

AP: Methodology design, Data collection, Data processing, analysis, & visualization, Writing –original draft & revisions. HP: Supervision, Resources, Methodology design, Data collection. AM: Methodology design, Data collection.

**Competing interests**

Some authors are members of the editorial board of *The Cryosphere*. The peer-review process was guided by an independent editor, and the authors have also no other competing interests to declare.

## Acknowledgments

The authors respectfully acknowledge that this research was conducted within the Chief Drygeese territory on the traditional
land of the Yellowknives Dene First Nation. The authors are grateful to the Indigenous Peoples for allowing the opportunity to learn and conduct field work on their lands. This research is supported by Government of Northwest Territories, Environment and Natural Resources, Cumulative Impact Monitoring Program (CIMP-212), Natural Sciences and Engineering Research Council of Canada (NSERC) Canada Research Chair and Discovery Grant to H. Kheyrollah Pour, Canada Excellent Research Chair-Global Water Futures (CERC-GWF), and Polar Knowledge Canada Northern Scientific Training Program
(NSTP). The authors also wish to acknowledge Tate Meehan for providing the initial code and insight for data processing.

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
