# Peer review of "Mapping snow depth over lake ice in Canada's sub-arctic using ground-penetrating radar"

_The Cryosphere, 2022_

## Author Comment (AC1)

**Comment on tc-2022-193**
**Response to Referee #1**
* * *
Referee comment on "Mapping snow depth over lake ice in Canada's sub-arctic using ground-penetrating radar" by Alicia Pouw et al., Cryosphere Discussion, https://doi.org/10.5194/tc-2022-193, 2022
* * *
The snow cover on lake ice is of great significance for the growth and decay of lake ice, lake climatology, limnological hydrology, and lake ecology. It is a positive work to develop a new technology based on the ground penetrating radar to quickly obtain the snow depth over large lake-ice areas. Based on the observation system, the authors carried out observation experiments in four lakes in the Canadian sub-Arctic region, proving the applicability and application value of the observation method, especially proving that the observation ability for the shallow snow layer over the ice surface. Thus, it is a method worth popularizing. The obtained data of large-scale snow observation can be further applied to the numerical simulation of lake ice and limnological hydrological processes, to evaluate the impact of snow and lake ice layers on the ecological environment of frozen lakes, and to evaluate the satellite remote sensing products of snow over the lakes. The paper is well written and structured, the method description is appropriate, the data analysis is basically sufficient, and the conclusion is clear, so it is a research work worth publishing in the TC. However, there are still some problems in the current expressions. It is mainly about the physical analysis of some data statistics results, and the impact of destruction of snowmobile track for natural snow surface on the observation data. Therefore, I recommend that the paper can be considered for publication only after a few minor revisions.

We are thankful to the reviewer, and we appreciate their suggestions and valuable comments for improving the manuscript. We have addressed or responded to all comments to improve the quality of this manuscript. Below, we provide the answers to the comments and questions raised by the reviewer. For convenience, comments from this Reviewer are provided in black text. Responses to each comment are provided in blue text.

General:

- Some statistical results based on observation data lack the analysis of potential physical mechanisms, for example, the difference of snow depth, density, relevant length in various lakes.

  Thank you for this comment. We have added an additional figure (Figure 11) to explore the snow depth and density changes from early winter season and late winter season on Landing Lake, where these measurements are collected. Additionally, we have added the following text to the results (Section 4.4: Early vs. late winter season). Additionally, we have added additional discussion on the relevant lengths for each lake which are discussed in more depth in coming comments.

[Figure]

**Figure 11:** Maps of Landing Lake snow depth (top) and density (bottom) in (a) December, (b) March, and (c) the difference between the two were created using IDWs of the GPR-derived snow depth and the in situ snow density observations.

**Lines 301-309:** "In comparing the difference in snow depth and snow density over the winter season, Figure 11 shows IDW 1-m snow depth maps and snow density maps (created using the in situ observations). The snow density from early season to late winter season increased between 10 to 80 kg/m$^3$, while the snow depth increased in areas by 18 to 28 cm. There were no surveyed areas on the lake that experienced a decrease in snow density or depth based on the two field sampling dates. Areas with a shallower snowpack in December 2021 saw the largest increase in snow depth by March 2022 (R$^2$ = 0.57), which agrees with the decrease in snow depth variability noted in Figure 9 by the correlation lengths. Additionally, the largest increase in density from early to late winter season occurred closest to the shoreline. More densification occurred on areas that were less dense than areas that had a higher density in December 2021 by March 2022 (R$^2$ = 0.59). In exploring the change in snowpack over the winter season, we found no spatial relationship between changes in the depth and density across the area surveyed on Landing Lake."

- The author said that snowmobile and sled rolling will increase the snow density and reduce the snow depth to a certain extent. The two impacts can offset each other, so their impacts are not significant. My suggestion here is whether you can further analyze the difference of the impact on thick and thin snow layers, on new and old snow layers, as well as on the snow accumulated in early December and the snow accumulated in late winter.

We appreciate your comment. We did revise the manuscript to discuss the crossover locations. We included discussion on the impact on thick and thin snow layers (or in terms of TWT – smaller and larger TWT) and for early and late winter snow accumulation. Following text is added to the revised manuscript:

**Lines 377-392:** "The analysis showed that no correction is required for compaction caused by the GPR sled. In considering the crossover locations (n = 533) on each of the lakes, we assessed the difference in TWT between the initial pass and the second pass and found that the average TWT difference was 0.02 ± 0.31 ns. Given the average velocity of 0.26 m/ns for the four lakes, and applying the one-quarter wavelength

Rayleigh criterion, the uncertainty of the TWT picks is approximately three samples (~0.3 ns). Therefore, the average TWT difference at crossover locations is within our uncertainty estimates of the TWT picks. In further exploring the change in TWT from the initial pass to the second, 56 % of the observations show the TWT for the second crossover to be larger than the initial. We found that shallower snow depths (or smaller TWTs) resulted in a decrease in travel time for the second pass, while deeper snow depths (or larger TWTs) showed an increase for the second pass for both early ($R^2$ = 0.30, $p < 0.05$) and late winter season ($R^2$ = 0.46, $p < 0.05$). However, these trends do not show dependency on the total snow depth accumulated throughout the winter season, as the average crossover differences of the data collections for early and late seasons (shallow and deep snow depths) are unbiased. Overall, although there is a change in density on the sled track ($\rho$ sled = 340 ± 20 kg/m3) compared to the density of the fresh snow (Table 2), the effects of a decrease in depth and increase in density under compaction from the snowmachine are naturally compensated and were confirmed with the crossover location TWT differences. The snow depth was measured at 1.5 cm less on average by using the density of the sled track for depth estimation rather than fresh snow density. Therefore, the effect on GPR derived snow depth is minimal because minimal snow mass was lost."

- This study presents observation data obtained from one winter. Although the data spatial coverage is relatively large, there is still a lack of data representativeness. Therefore, it is suggested to increase the discussion of data representativeness obtained from the observed winter. How does the snow accumulation on land compare with previous years? What is the difference of the atmospheric precipitation, temperature and other parameters in the winter of the observation related to the climatology? etc. Through such comparison, the application value of observation data can be enhanced.

Thank you for this comment. To further improve the data representativeness, we have added the following lines to the revised manuscript:

**Lines 320-338:** "Lake freeze up for small lakes surrounding Yellowknife generally occurs during October, however, lake freeze up was reported to occur later this year compared to the 2018 to 2020 seasons based on Yellowknife's snowmobile association data. October air temperatures reported at the Yellowknife weather station showed a mean temperature increase of 4.4°C between 2020 (-1.85°C) and 2021 (2.6°C), and a 3.18°C increase when comparing to the 5-year and 10-year October mean air temperatures. Within the 2021 to 2022 water year, ~ 75 cm of snowfall was reported by the Yellowknife weather station, accounting for 46 % of total annual precipitation. In comparing the snowfall to previous years, the 2021 to 2022 water year experienced 20 % less snowfall than the 2020 to 2021 water year (~93 cm and 76 % of total precipitation). In the past 5 to 10 years, on average, 40 to 45 % more snowfall was reported compared to the 2021 to 2022 year. The timing and amount of snowfall will influence the lake ice composition, thickness, and phenology. Larger amounts of snow accumulation on thin, early season lake ice with reduced buoyancy will create leads and cause overflow, which increases the likelihood of snow ice growth. Thin and patchy snow ice (0 – 2 cm) was observed on the lake ice surface during the December and March field campaigns, making up 0 % to 6 % of the lake ice composition. Based on observations recorded up until March 2022, scarce amounts of snow ice were present, which suggests that minimal overflow occurred throughout the winter season on these four lakes prior to the beginning of ice break up.

In December 2021 and March 2022, the lakes consistently showed a shallower snowpack on average (Table 3) than snow on the ground (Figure 2) reported at the nearby Yellowknife weather station. The lakes measured at an average of 24 % to 29 % less snow than measured over land in December 2021, and 15 % less in March 2022. Thus,

assuming snow depths measured on land as an input to lake ice models will overestimate lake snow depth by a seasonally dependent factor and impact the modeled ice thickness (Kheyrollah Pour et al., 2017).”

Special comments:

- Line 15 “~9 cm spatial resolution along transects” 9-cm is the sampling resolution, not the data resolution, because you have not considered the footprint of observation. Therefore, it is recommended to further analyze the observation footprint of single observation.

  We thank the reviewer for this comment. We agree with the comment and modified the text in the revised manuscript from “spatial resolution” to “sampling resolution” when referencing the GPR trace spacing. In addition, we added the following text to the revised manuscript in the methodology (Section 3.1: GPR data acquisition)

  **Lines 143-148:** “The average footprint of each collected trace on all four lakes in December was 19 cm, and 30 cm in March on Landing Lake based on the diameter of the first Fresnel zone (Fediuk et al., 2022). In considering the ~9 cm trace spacing to the footprint of each trace, the data results in over 50 % overlap. The vertical imaging resolution was estimated at 6.5 cm on average across all four lakes based on the one-quarter wavelength Rayleigh criteria using the 1000 MHz sensor (Kallweit and Wood, 1982), which has a vertical sampling interval of 0.1 ns.”

- Introduction: The application of observation data of snow over the lake ice cannot only focus on the developing of lake ice numerical model, but also be applied to lake ice phenology (e.g., Lei et al., 2012), lake ecology and other fields. The description of research background should be more comprehensive in the introduction.

  Ref.: Lei R, Leppäranta M, Cheng B, et al. Changes in ice-season characteristics of a European Arctic lake from 1964 to 2008. Climatic change, 2012, 115(3-4): 725-739.

  We thank the reviewer for this comment and the additional reference provided. The description of research background has been expanded on in the introduction for the revised manuscript as suggested.

  **Lines 34-59:** “A challenge to measuring lake snow is the inconsistent snow thickness across the lake. Snow redistributed by wind commonly deposits on the leeward side of topographic features. Snow accumulation on lake ice surrounding these features (i.e., pressure ridges) leads to the formation of snowdrifts. Additionally, snow dunes will form in areas of turbulent winds on relatively level ice surfaces (Liston et al., 2018). The formation of snowdrifts and snow dunes creates a heterogenous snow thickness across the ice surface. The uneven snow depth distribution leads to spatial variability in the lake ice thickness due to the increase in heat transfer through the snow for areas of shallow snow (assuming a constant thermal conductivity). Micro-topographic snow features impact the ice mass balance and must be considered when evaluating local and regional energy balances and fluxes (Sturm et al., 2002).

  Snow and lake ice are sensitive to a change in daily air temperature (Rafat et al., 2022). As warming is occurring in Northern Canada at twice the global rate and is expected to continue to increase (Zhang et al., 2019), a change in the surface-atmosphere energy balance will directly affect snow and lake ice conditions (Brown and Duguay, 2010). Within the changing climate, a change in snow cover (Brown et al., 2021; Mudryk et al., 2017), lake ice phenology (timing of ice formation and break up; Magnuson et al., 2000; Lei et al., 2012; Benson et al., 2011), and ice thickness and composition (Kholoptsev et al., 2021) are being observed. Spatial and temporal observations of lake snow and ice can be indicators to changes in climatic variables. Later freeze up and

earlier break-up of ice cover leads to an extended open-season and can influence the lake surface water temperatures (i.e., Woolway et al., 2021), affecting the lake biogeochemical processes (e.g., Adrian et al., 2009; Jeppesen et al., 2014). Additionally, northern communities rely on lake ice for cultural and recreational use, and as a source of transportation through ice roads (Knoll et al., 2019). Ice roads allow travel to neighbouring communities and alternative access to goods and supplies (instead of transport via airplane). With warming projected to increase, it can be expected that the safety of ice roads and operational duration will be affected (Stephenson et al., 2011; Mullan et al., 2021). As the presence of snow over lake ice directly affects ice thickness, measuring snow depth on lake ice is crucial for lake modelling and ice thickness estimation on a regional scale. Previous studies by Kheyrollah Pour et al (2017) show that accurate snow depth observations over lake ice can significantly improve the thermodynamic lake ice models.

Improving snow depth observations and retrieving an accurate higher spatial resolution snow depth is essential for hydrological, limnological, and lake ice studies (Lei et al., 2012; Kheyrollah Pour et al., 2017; Marsh et al., 2020; Li et al., 2022)."

- Line 46 "Daily snow depths are reported across Canada using instruments, such as." As you mentioned later, the SnowHydro Magnaprobe is a common method for snow depth measurement. Therefore, it should be introduced in introduction, and its advantages and disadvantages should be described, such as manual operation, which is not conducive to obtaining a wide range of snow depth observation data.

Thanks for the comment. We have added the text in the revised manuscript introducing the magnaprobe and its advantages and disadvantages as follows:

**Lines 66-77:** "Currently, retrieving accurate snow depth observations over lake ice and mapping the spatial distribution and heterogeneity of snow over ice is challenging because of the limited support of point measurements using contemporary methods, such as a ruler and notebook or automatic snow depth probe. An automatic snow depth probe, such as the magnaprobe, is equipped with a metal rod probe that penetrates the snowpack to the ice surface and a sliding basket that sits on the surface of the snow, recording the snow depth and spatial location when manually placed in position (Sturm and Holmgren, 2018). The magnaprobe records the snow depth accuracy with errors ranging from near zero for hard bases to +5 cm. The Wide Area Augmentation System-enabled GPS provides a position accurate to ±2.5 m. The advantages of using a magnaprobe is the increase in speed with which a depth and position measurement can be obtained by a factor of 10 compared to measuring with a traditional ruler probe and writing down the results. The highest boost in snow depth measurement efficiency occurs when the distance between measuring locations is kept relatively small (<10 m). The snow depth probe has been commonly utilized for validation of remote sensing techniques (*i.e.,* McGrath et al., 2019; Walker et al., 2020), however, due to the limited spatial coverage that current methods pose; it is not logistically feasible to measure the snow depth on lake-wide scales."

- Line 83 "It is expected that the wind fetch and shoreline vegetation affect the snow distribution", However, in the later data analysis, the impact of these two factors on different lakes has not been discussed enough.

We thank the reviewer for this comment. We agree with the comment and expand the discussion of how the wind fetch and shoreline vegetation were found to affect the snow distribution in the revised version of the manuscript as follows. First, we expand on the study area and add the following text (Section 2: Study area):

**Lines 111-117:** "These lakes are part of a turbulent wind field, as the wind direction and speed reported at the Yellowknife weather station vary rapidly. The most predominant winds in December and November came from the east (~27 %) and had an average wind speed of 9 km/h, with the strongest winds coming from the northeast (~15 %) reaching 33 km/h. Throughout January to March, the strongest winds came from the northwest (~22 %) reaching 37 km/h, but frequent winds came from the northeast in January (~22 %), northwest in February (~26 %) and northeast, east, and northwest in march (~21 %) travelling at 11 km/h on average, while very little winds were recorded from the south (~6 %) between October to March."

The following text is added in the results (Section 4.3: Snow depth mapping):

**Lines 270-273:** "The interpolated GPR-snow depths consistently show an increase in snow depth variability closer to the lake perimeter compared to areas farther from the shoreline and closer to the center of the lake. The snow depth on Finger Lake showed a decrease of ~2 cm per meter as the distance from the perimeter increased, however, this was not observed on the additional lakes."

The following text is added in the discussion to clarify on the differences between the lakes:

**Lines 339- 347:** "On relatively level ice surfaces and in turbulent wind fields, snow dunes are formed from snow redistributed by wind. The snow depth accumulation over the lakes varied but could be explained by the total snowfall (8 cm) with consideration to wind redistribution and compaction seen between December 7th (Landing-D Lake $\overline{h}_s$ = 12.76 cm) to December 14$^{th}$ (Vee Lake $\overline{h}_s$ = 16.06 cm). During both field campaigns there was evidence of snow dunes present across the lakes. This study explored the distribution of snow over each lake (Figure 9), which showed local-scale variability of snow depths from redistribution of the snow across all the lakes (correlation lengths between 6–19 m). We used semi-variogram analyses to determine the horizontal spacing of the snow dunes and found Long Lake to have the shortest correlation length (6.42 m). On Landing Lake, we saw an increase in correlation length throughout the winter season from ~7 m to ~19 m. The inferred variability length-scales are similarly supported in the literature, reporting correlation lengths from 5 to 20 m (Gunn et al., 2021a; Sturm and Liston, 2003)."

**Lines 361-369:** "The snow distribution over lake ice is known to be affected by wind and surrounding vegetation (Adams, 1976a). In this study we found weak relationships between the lake snow depth and distance to shoreline perimeter. On Finger Lake where we have more complete coverage of the lake, we found the snow depth to decline ~2 cm per meter from the shoreline to the centre of the lake but found no change on the additional three lakes. We believe this could be due to the lack of data representativeness around the shoreline and the difficulty associated with maneuvering the snowmobile in the deep, lighter snow at slow speeds, or the turbulent winds affecting which shoreline the snow will be distributed along. Winds reported at the Yellowknife weather station reached speeds above the ~14 to 39 km/h threshold required to transport snow (Li and Pomeroy, 1997), however, with the majority of strong winds coming from the northeast and northwest, our lack of data on the southern perimeter on each lake may also affect our findings."

- Table 2: Could you explain why the Long Lake has a relative large snow density compared to other lakes?

We believe the relatively large snow density on Long Lake is due to the surface area of the lake compared to the additional other three lakes. The lake has a larger wind fetch due to the shape and the location along to the highway. We have added the following clarification in the discussion of the revised manuscript:

**Lines 348-353:** "In comparing the spatial snow depth variability across the four lakes, we believe the physical characteristics of Long Lake explain the reduced correlation length in comparison to the three additional lakes. Long Lake has the largest surface area to perimeter ratio and spans ~3 km northwest to southeast. Therefore, Long Lake exhibits the largest wind fetch area compared to the additional three study areas and can explain the higher snow density compared to the other lakes. While on Landing Lake, both the snow depth and density increased over the season, however, to determine the reason for the decrease in snow depth variability from December to March, more frequent sampling dates would have to occur between early and late season."

- Line 199 "area = 4 ha" ha is not the International Standard Unit.

  Done. We have switched to state area as 0.04 $km^2$. (4 ha). Thank you.

- Figure 5: In fact, there are multiple intersections in the observation transects for all lakes, which means that there should be two observations at these intersections. In order to explain the stability of the observation and retrieval results, it is necessary to compare the repeated observation results obtained from these measurement intersections.

  Thanks for the comment. To address this comment, we added the text outlined in the above general comment (**Lines 377-392**).

  **Lines 377-392:** "The analysis showed that no correction is required for compaction caused by the GPR sled. In considering the crossover locations (n = 533) on each of the lakes, we assessed the difference in TWT between the initial pass and the second pass and found that the average TWT difference was $0.02 \pm 0.31$ ns. Given the average velocity of 0.26 m/ns for the four lakes, and applying the one-quarter wavelength Rayleigh criterion, the uncertainty of the TWT picks is approximately three samples (~0.3 ns). Therefore, the average TWT difference at crossover locations is within our uncertainty estimates of the TWT picks. In further exploring the change in TWT from the initial pass to the second, 56 % of the observations show the TWT for the second crossover to be larger than the initial. We found that shallower snow depths (or smaller TWTs) resulted in a decrease in travel time for the second pass, while deeper snow depths (or larger TWTs) showed an increase for the second pass for both early ($R^2 = 0.30$, $p < 0.05$) and late winter season ($R^2 = 0.46$, $p < 0.05$). However, these trends do not show dependency on the total snow depth accumulated throughout the winter season, as the average crossover differences of the data collections for early and late seasons (shallow and deep snow depths) are unbiased. Overall, although there is a change in density on the sled track ($\bar{\rho}_{sled} = 340 \pm 20$ $kg/m^3$) compared to the density of the fresh snow (Table 2), the effects of a decrease in depth and increase in density under compaction from the snowmachine are naturally compensated and were confirmed with the crossover location TWT differences. The snow depth was measured at 1.5 cm less on average by using the density of the sled track for depth estimation rather than fresh snow density. Therefore, the effect on GPR derived snow depth is minimal because minimal snow mass was lost."

- Lines 222, 225 "Long Lake showed the lowest agreement", "with Vee Lake being the most accurate": Corresponding to such measurement difference, some physical explanations are required.

We appreciate the reviewer's comment. We expect the snow depth variability on long lake to vary on a shorter length scale due to the surface area, shape, and location of Long Lake compared to the other three additional lakes, which attributes to the accuracy of the derived-snow depths with using a 6m radius. Similarly, Vee lake has higher agreement most likely due to the deeper snowpack (more accurate to derive deeper snowpack with GPR) and also Vee lake has the largest correlation length, meaning there is less variability in the snow depth within the 6 m radius used to derive the snow depth. To address this further, we have added to the following text in the discussion:

**Lines 343-360:** "We used semi-variogram analyses to determine the horizontal spacing of the snow dunes and found Long Lake to have the shortest correlation length (6.42 m). On Landing Lake, we observed an increase in correlation length throughout the winter season from ~7 m to ~19 m. The inferred variability length-scales are similarly supported in the literature, reporting correlation lengths from 5 to 20 m (Gunn et al., 2021a; Sturm and Liston, 2003).

In comparing the spatial snow depth variability across the four lakes, we believe the physical characteristics of Long Lake explain the reduced correlation length in comparison to the three additional lakes. Long Lake has the largest surface area to perimeter ratio and spans ~3 km northwest to southeast. Therefore, Long Lake exhibits the largest wind fetch area compared to the additional three study areas and can explain the higher snow density compared to the other lakes. While on Landing Lake, both the snow depth and density increased over the season, however, to determine the reason for the decrease in snow depth variability from December to March, more frequent sampling dates would have to occur between early and late season.

We believe the lower accuracy in GPR-derived snow depths on Long Lake (±11 %) could be attributed to using a radius to compare the derived and in situ snow depths that was approximately the same magnitude as the length scale of snow depth variability. Vee Lake had the highest accuracy (±6 %) in deriving the snow depth and the largest correlation length (~11 m) in December 2021. The greatest accuracy (± 5 %) was found during the late season on Landing-M Lake which also found to have the largest correlation length (~19 m). Therefore, the snow depth variability within 6 m was less on Vee Lake and Landing Lake than on Long Lake. Overall, we may expect the accuracy to increase by improving the spatial location of the in situ snow depth measurements and sampling more frequently within the length scales of each lake."

- relative error = 11.04 %, and other somewhere: For relative errors, it is not necessary to retain two decimal places, because the accuracy of the evaluation cannot reach this level.

  Thank you. We have adjusted the relative error to whole numbers in the revised version of the manuscript, updating Table 4, as well as throughout the results section.

- Line 240 "However, the relative error was improved on Landing-M Lake with a deeper snowpack (5.33 %) than that of Landing-D Lake (8.06 %). During the later season, the GPR could derive the minimum snow depths seen on Landing Lake, as opposed to that in the early season, where" Some further explanation is needed here, not only to give the data results.

  We appreciate the reviewer's comment. The following text is added to the revised manuscript:

  **Lines 294-296:** "However, the relative error was improved on Landing-M Lake with a deeper snowpack (5 %) than that of Landing-D Lake (8 %). The GPR could derive the minimum snow depths seen on Landing Lake during the later season, as opposed to that

in the early season, where the GPR-derived snow depth could not capture the shallowest snow area (4.5 – 10 cm)."

We also mention later in the discussion that a 7cm threshold has to be applied due to the wavelength of the 1000 MHz with the snow (vertical sampling resolution).

**Lines 372-376:** "Overall, the results of this study showed that a 7 cm threshold exists as a limitation of deriving shallow snow depth from GPR TWT using the 1000 MHz sensor. Showing similar agreement with previous studies (Pfaffhuber et al., 2017), the in situ observations below 7 cm were not considered in the validation analysis. During the March 2022 campaign, seldomly snow depth was observed below 25 cm, meaning the vertical imaging resolution of 6.5 cm for the 1000 MHz sensor did not limit our data acquisition."

---

## Author Comment (AC2)

**Comment on tc-2022-193**
**Response to Referee #2**
* * *
Referee comment on "Mapping snow depth over lake ice in Canada's sub-arctic using ground-penetrating radar" by Alicia Pouw et al., Cryosphere Discussion, https://doi.org/10.5194/tc-2022-193, 2022
* * *
The manuscript "Mapping snow depth over lake ice in Canada's sub-arctic using ground-penetrating radar" presents a study that takes a commonly used method (GPR) and applies it to snow on lake ice. The study is able to cover great distances with high spatial resolution of observations and compare GPR depth estimates to manual depth measurements with a Magnaprobe. The GPR method resulted in an estimated RMSE of 1.58 cm with a mean bias of -0.01 cm during the early season, and RMSE of 2.86 cm and bias of 0.41 cm later in the season.

Overall, the authors produce a very nice dataset that can be used for modeling efforts and potentially remote sensing validation. However, I do not see anything that justifies this study to be at the level of a "Research Article" in The Cryosphere. Again, this is a great dataset but a more robust analysis of data would need to be presented to be a research article, in my opinion. As it is I think it a great "data paper" or potentially a "technical note" type of manuscript. Unfortunately, The Cryosphere does not publish these types of papers so I recommend either submitting to another journal pretty much as is, or providing further quantitative analysis to bump it up to being a full research article. The variograms are a great start, but I think more information on the spatial variability of the snow on lake ice could be good to include. This could include for example: directional variograms to investigate isotropy, variability or depth as a function of distance to shore or distance to islands, does topography of the shore or presence of trees impact anything. I think that the authors started to go down this route with Figure 9 but it needs to continue for more statistical quantifications, in my opinion.

One reason further analysis would be necessary is because the authors did not develop any new tools advance any of the methods to collect the data. Further minor comments are listed below by line number.

We are thankful to the reviewer, and we appreciate their suggestions and valuable comments. Below, we provide the answers to the comments and questions raised by the reviewer. For convenience, comments from this Reviewer are provided in black text. Responses to each comment are provided in blue text.

General:

We do appreciate the reviewer's comments and suggestions. However, we do not fully agree with the comment related to the lack of introducing a new tool to advance any of the methods to collect the snow depth data on lake ice. This research article presents an automated approach to derive spatial shallow snow depth observations over lake ice using GPR, which, from our knowledge, has not previously been done and constantly reported that GPR is not able to retrieve lake snow depth and distribution. This research article provides a new inside of how GPR can retrieve even shallow

snow depth on lake ice using a fully automated approach. Although GPR is commonly utilized for snow on land, this does not highlight the ability to discern shallow snow depths more specifically over lakes with lack of ice elevation. It has also been used on lake ice to derive the thickness of the ice, however the snow has always been ignored or cleared before the measurements. In addition, the shallower snow depths are difficult to derive due to the direct wave conflicting with the snow-ice interface, however, we present a method that the GPR is capable of decerning that. Additionally, the GPR acquisitions are fully post-processed in MATLAB and does not require use of any additional software. This includes the signal processing and the TWT picking algorithm. This decreases the required time for post-processing radargrams and using manual or semi-automatic picking algorithms that have been commonly used for GPR data in the past.

Current methods available for observing snow depth over lake ice, as outlined in the manuscript, include a ruler and notebook with a GPS – which requires a lot of unnecessary time recording each observation individually, or an automated snow depth probe, which while great for validation, is not the most appropriate option for covering the lake spatially in a timely manner. The method presented within this manuscript is successful in producing 1) a fully automated post-processing workflow for all data processing/ analysis steps, 2) is successful with shallow lake snow and 3) can collect large spatial data sets in a limited time.

Here, we modified the revised version of the manuscript as suggested, including the difference in snow depth and density maps from early season to late winter season, as well as the discussion on variability/depth as a function of distance to shore. We have added an additional figure (Figure 11) to explore the snow depth and density changes from early winter season and late winter season on Landing Lake. Additionally, we have added the following text to the results (Section 4.4: Early versus late winter season):

**Lines 301-309**: "In comparing the difference in snow depth and snow density over the winter season, Figure 11 shows IDW 1-m snow depth maps and snow density maps (created using the in situ observations). The snow density from early season to late winter season increased between 10 to 80 kg/m$^3$, while the snow depth increased in areas by 18 to 28 cm. There were no surveyed areas on the lake that experienced a decrease in snow density or depth based on the two field sampling dates. Areas with a shallower snowpack in December 2021 saw the largest increase in snow depth by March 2022 (R$^2$ = 0.57), which agrees with the decrease in snow depth variability noted in Figure 9. Additionally, the largest increase in density from early to late winter season occurred closest to the shoreline. More densification occurred on areas that were less dense than areas that had a higher density in December 2021 by March 2022 (R$^2$ = 0.59). In exploring the change in snowpack over the winter season, we found no spatial relationship between changes in the depth and density across the area surveyed on Landing Lake."

Following text is added on to the discussion of variability/depth as a function of distance to shore in the manuscript (Section 4.3 Snow depth mapping)

**Lines 270-278**: "The interpolated GPR-snow depths consistently show an increase in snow depth variability closer to the lake perimeter compared to areas farther from the shoreline and closer to the center of the lake. The snow depth on Finger Lake showed a decrease of ~2 cm per meter as the distance from the perimeter increased, however, this was not observed on the additional lakes. Transect profiles (Figure 8) created over the 1 m resolution snow depth maps show an example of

the variability in snow depth across each lake. The spatial correlations of the 1 m resolution snow depths from the GPR transects were estimated using an experimental semi-variogram that was fit using an exponential model (Figure 9). The largest correlation length was observed on Vee Lake (11.25 m) in December 2021, and Landing-M Lake (18.18 m) overall. The correlation length on Landing-D Lake in the early season was measured at ~10 m less than that of the late winter season, while Long Lake showed the smallest distance, at 6.42m over the largest spatial area."

Following text is added on discussion (Section 5: Discussion):

**Lines 361-369:** "The snow distribution over lake ice is known to be affected by wind and surrounding vegetation (Adams, 1976a). In this study we found weak relationships between the lake snow depth and distance to shoreline perimeter. On Finger Lake where we have more complete coverage of the lake, we found the snow depth to decline ~2 cm per meter from the shoreline to the centre of the lake but found no change on the additional three lakes. We believe this could be due to the lack of data representativeness around the shoreline and the difficulty associated with maneuvering the snowmobile in the deep, lighter snow at slow speeds, or the turbulent winds affecting which shoreline the snow will be distributed along. Winds reported at the Yellowknife weather station reached speeds above the ~14 to 39 km/h threshold required to transport snow (Li and Pomeroy, 1997), however, with the majority of strong winds coming from the northeast and northwest, our lack of data on the southern perimeter on each lake may also affect our findings."

Specific comments:

- 15: 9 cm spatial resolution is the spacing between traces, but after you aggregate the data it is a 1 m raster correct? This is the resolution of the data that should be reported and also incorporates the footprint of observations.

    We thank the reviewer for this comment. We agree with the comment and modified the text in the revised manuscript as follows:

    **Lines 143-148:** "The average footprint of each collected trace on all four lakes in December was 19 cm, and 30 cm in March on Landing Lake based on the diameter of the first Fresnel zone (Fediuk et al., 2022). In considering the ~9 cm trace spacing to the footprint of each trace, the data results in over 50% overlap. The vertical imaging resolution was estimated at 6.5 cm on average across all four lakes based on the one-quarter wavelength Rayleigh criteria using the 1000 MHz sensor (Kallweit and Wood, 1982), which has a vertical sampling interval of 0.1 ns."

- 115: "was" should be "were"

    This change has been made. Thank you!

- 158: How was the Wong et al. algorithm applied? Matlab? Python? Please specify.

    Thank you, we have added the following text:

**Line 181:** "All post-processing of the radargrams was conducted in MATLAB."

- 184-190: How much variability occurred over the 6 m. It seems to me that by choosing only values that closely match one would underestimate the magnitude of the error/bias. As it is written, I do not see a justification for this current method and think the authors should use all values within the 6 m range to calculate the comparison metrics.

  Thank you for this comment. The reason we don't use the entirety of the data within the 6 m is because of the variability across the lakes for short distances. For example, the correlation length for long lake shows that the lake variability is on the same scale as the 6 m radius, which agrees with previously published articles on lake snow (Gunn et al., 2021; Sturm and Liston, 2003). In discussing the variability within the 6 m, one standard deviation of the derived snow depths within a 6 m distance is between 2.1 cm to 3.8 cm (Finger = 3.4 cm, Long = 3.8 cm, Vee = 3.2 cm, Landing-D = 3.1 cm, and Landing-M = 2.1 cm).

- 200: what is meant by "closed-off areas"

  Thank you for the comment. The closed-off areas were referring to areas surrounded closely by the perimeter with but not as wide open as the centre of the lake. We have reworded to read as follows:

  **Lines 234-235:** "The entirety of Finger Lake (area = 0.04 km$^2$) was traversed on December 9$^{th}$, where the deepest snow depths were observed along shorelines (max = 24.83 cm), compared to the open stretch of the lake (min = 6.53 cm)."

- 236: Given such low density values, I am not sure that teh Kovacs equation is appropriate. Kovacs was developed for much denser firn. Di Paolo et al. (2018) and Webb et al. (2021) could be good references for a more appropriate equation.

  Thank you for this comment. Within our analysis we use Kovacs et al. (1995) equation to derive the permittivity as we found minimal variability in deriving the relative permittivity using different equations for shallower lake snow. Additionally, Di Paolo et al. (2018) shows in Figure 1 that in comparing 17 different empirical formulas to calculate permittivity, there is not as much variability for lower densities than there is for higher density snowpacks (i.e., 300kg/m$^3$ to 550 kg/m$^3$). In addition, with testing different empirical relationships to calculate the permittivity (i.e., Di Paolo et al., 2018; Webb et al., 2021; Stein et al., 1997; Tiuri et al., 1984), there is very slight differences in the dielectric constant, if any at all. Which when further used to derive the snow depth, the largest difference in accuracy from comparing the GPR-derived snow depth to the in situ was found with Webb et al. (2021) with an R$^2$ =0.61, MAE = 1.08 cm, RMSE = 1.61 cm, Bias = 0.47 cm on average for the four lakes. Within the study we report an R$^2$ =0.63, MAE = 1.05 cm, RMSE = 1.58 cm, Bias = 0.01 cm using the Kovacs et al. (1995) equation, showing millimetre differences. These results from the Kovacs et al. (1995) method is however identical to using the Tiuri et al. (1984), Robin et al. (1969), Robin (1975), Frolov & Macheeret (1999), and very similar to Stein (1997) equation (R$^2$ =0.62, MAE = 1.06 cm, RMSE = 1.58 cm, Bias = 0.08 cm). To bring more attention to our decision we have added the following text to the revised manuscript in the methodology (Section 3.3.4):

**Lines 201-206**: "To determine the wave speed of the radar signal traveling through the snow, the relative permittivity was calculated. There are several empirical equations available for deriving the relative permittivity from snow density. Previous work (i.e., Di Paolo et al., 2018; Webb et al., 2021) found there is significant variability between these equations for larger snow densities, however, based on the snow densities presented within this study, there is minimal variability between equations. Therefore, the Kovacs et al. (1995) equation is used to calculate the relative permittivity."

**Lines 393-407:** "Lake snow is not well characterized in the various dielectric permittivity models used for wave speed estimation. In this study we found the snow depth retrieval is weakly dependent on the choice of empirical equation used to derive the snow depth from density. Within our analysis we used the Kovacs et al. (1995) equation to derive the permittivity. In addition, we also tested different empirical relationships to calculate the permittivity (i.e., Robin et al., 1969; Robin, 1975; Tiuri et al., 1984; Stein et al., 1997; Frolov and Macheret, 1999, Webb et al., 2021) and found very slight differences in the dielectric constant, if any at all. The results ($\overline{\varepsilon_r}$ = 1.37) from the Kovacs et al. (1995) method are identical to using the Robin et al. (1969), Robin (1975), Tiuri et al. (1984), Frolov and Macheret (1999), and very similar to Stein et al. (1997) equation ( $\overline{\varepsilon_r}$ = 1.34), with the largest difference using the Webb et al. (2021) equation ($\overline{\varepsilon_r}$ = 1.29). In exploring the permittivity for the snow densities presented within this study (175 kg/m$^3$ to 245 kg/m$^3$), the numerous empirical relationships result in very similar permittivity's for these lower densities and sub-millimetre differences in the snow depth accuracy statistics (not shown). Di Paolo et al. (2018) shows in comparing 19 different empirical formulas to calculate permittivity, there is less variability for lower densities than there is for higher density snowpacks (i.e., 300 kg/m$^3$ to 550 kg/m$^3$). Di Polo et al. find that Robin Lake snow has generally been reported to be shallower and less dense than snow types used to parameterize these models. However, based on the agreement among models and the limited representation for a model based on lake snow observations we have sided with the Kovacs et al. (1995) equation."

references:

Di Paolo, F.; Cosciotti, B.; Lauro, S.E.; Mattei, E.; Pettinelli, E. Dry snow permittivity evaluation from density: A critical review. In Proceedings of the 2018 17th International Conference on Ground Penetrating Radar (GPR), Rapperswil, Switzerland, 18–21 June 2018; pp. 1–5

Webb, R.W.; Marziliano, A.; McGrath, D.; Bonnell, R.; Meehan, T.G.; Vuyovich, C.; Marshall, H.-P. In Situ Determination of Dry and Wet Snow Permittivity: Improving Equations for Low Frequency Radar Applications. Remote Sens. 2021, 13, 4617. https://doi.org/10.3390/rs13224617

These comments are meant to be constructive. I think this is an excellent dataset and good work.

---

## Author Comment (AC3)

**Comment on tc-2022-193**
**Response to Community Comment #1**
* * *
Community comment on "Mapping snow depth over lake ice in Canada's sub-arctic using ground-penetrating radar" by Alicia Pouw et al., Cryosphere Discussion, https://doi.org/10.5194/tc-2022-193, 2022
* * *
Snow accumulation on lake ice can have a significant impact on the evolution of the ice cover, particularly as wind-driven forces can cause significant spatial variation in the distribution of snow on the ice cover. This is more pronounced in areas of low snowfall, where there are surface conditions of both bare ice and snow crossings, which are critical to the overall heat content of the lake ice. It is currently difficult to quantify precisely the spatial distribution of snow thickness, and shallow snow cover is also a dominant natural phenomenon in many mid-latitude regions. This technique allows rapid access to snow depths over large areas of lake ice as opposed to traditional manual measurements and fixed-point automated observations. It is a valuable tool for estimating and analysing the thermal balance of the ice surface over the entire lake ice and for gaining a clearer understanding of the physical processes involved in snow redistribution.

We are thankful to the valuable comments and questions towards the manuscript. We have responded to all comments to improve the manuscript. Below, we provide the answers to the comments and questions raised. For convenience, the community comments are provided in black text. Responses to each comments/questions are provided in blue text.

Some questions are as follows:

- The rolling of snowmobile and sled compacts the snow, can the reduction in depth and the increase in density be completely offset? This is because in the case of the study where the snow is deeper, the compaction does not act evenly across the snow layer resulting in an uneven increase in overall density. Would it be better if in the future the snowmobiles were to "push" the sleds instead of "pulling" them, or would it be better if they were to be carried by drones?

  Thank you for your question. In our revised manuscript we have added more discussion on the compaction of the snow caused by the sled. We looked at the crossover locations and compared the difference in TWT for the initial pass compared to the second, and found an average difference of $0.02 \pm 0.31$ ns. This aligns with the uncertainty of the TWT picks (~0.3 ns). In exploring the difference in TWT instead of a function of snow depth or density, we can assume the change in one parameter is compensated for by a change in the other, which agrees with McGrath et al. (2019). Additionally, in looking at the sensitivity in deriving snow depth with density, there is minimal impact on the GPR-derived snow depth with a change in density based on density observations recorded in the field.

  In the future, to further confirm this is the case, mounting the GPR on the front of the skidoo, hovering right on the snow surface would avoid compaction caused from the

snowmobile and sled that the GPR sits in. The problem of having a gap between the radar and the snow surface is detecting the air-snow and snow-ice interface within a short time window could present challenges. This would be more achievable in a deeper snowpack, however, would be challenging in shallow lake snow due to the reflections that would be caused from the air-snow interface, and the interference it would cause in decerning the snow-ice interface (due to the vertical imaging resolution) even with removal of the direct wave.

**Lines 377-392**: "The analysis showed that no correction is required for compaction caused by the GPR sled. In considering the crossover locations (n = 533) on each of the lakes, we assessed the difference in TWT between the initial pass and the second pass and found that the average TWT difference was $0.02 \pm 0.31$ ns. Given the average velocity of 0.26 m/ns for the four lakes, and applying the one-quarter wavelength Rayleigh criterion, the uncertainty of the TWT picks is approximately three samples (~0.3 ns). Therefore, the average TWT difference at crossover locations is within our uncertainty estimates of the TWT picks. In further exploring the change in TWT from the initial pass to the second, 56 % of the observations show the TWT for the second crossover to be larger than the initial. We found that shallower snow depths (or smaller TWTs) resulted in a decrease in travel time for the second pass, while deeper snow depths (or larger TWTs) showed an increase for the second pass for both early ($R^2 = 0.30$, $p < 0.05$) and late winter season ($R^2 = 0.46$, $p < 0.05$). However, these trends do not show dependency on the total snow depth accumulated throughout the winter season, as the average crossover differences of the data collections for early and late seasons (shallow and deep snow depths) are unbiased. Overall, although there is a change in density on the sled track ($\bar{\rho}$ sled = $340 \pm 20$ kg/m3) compared to the density of the fresh snow (Table 2), the effects of a decrease in depth and increase in density under compaction from the snowmachine are naturally compensated and were confirmed with the crossover location TWT differences. The snow depth was measured at 1.5 cm less on average by using the density of the sled track for depth estimation rather than fresh snow density. Therefore, the effect on GPR derived snow depth is minimal because minimal snow mass was lost."

 The authors obtained snow depth data with a large spatial coverage and also assessed the accuracy of the data. Consideration could be given to discussing this in the context of climatic background and terrain features to improve the potential application of the data. For example, is the variability in snow depth influenced by the wind speed and direction prior to measurement? Is the greater depth of snow on the banks due to the barrier effect of vegetation or bank slopes?

Thank you for these comments. We did add more analysis related to the snow depth distribution into the revised manuscript, where we considered the distance to the shoreline and discuss the micro-topographic snow features. We have also added an additional figure (Figure 11), looking at the difference in snow depth and density between the December 2021 and March 2022 field campaign.

[Figure]

**Figure 11:** Maps of Landing Lake snow depth (top) and density (bottom) in (a) December, (b) March and (c) the difference between the two were created using IDWs of the GPR-derived snow depth and the in situ snow density observations.

**Lines 301-309:** "In comparing the difference in snow depth and snow density over the winter season, Figure 11 shows IDW 1-m snow depth maps and snow density maps (created using the in situ observations). The snow density from early season to late winter season increased between 10 to 80 kg/m$^3$, while the snow depth increased in areas by 18 to 28 cm. There were no surveyed areas on the lake that experienced a decrease in snow density or depth based on the two field sampling dates. Areas with a shallower snowpack in December 2021 saw the largest increase in snow depth by March 2022 (R$^2$ = 0.57), which agrees with the decrease in snow depth variability noted in Figure 9 by the correlation lengths. Additionally, the largest increase in density from early to late winter season occurred closest to the shoreline. More densification occurred on areas that were less dense than areas that had a higher density in December 2021 by March 2022 (R$^2$ = 0.59). In exploring the change in snowpack over the winter season, we found no spatial relationship between changes in the depth and density across the area surveyed on Landing Lake."

**Lines 339- 347:** "On relatively level ice surfaces and in turbulent wind fields, snow dunes are formed from snow redistributed by wind. The snow depth accumulation over the lakes varied but could be explained by the total snowfall (8 cm) with consideration to wind redistribution and compaction seen between December 7th (Landing-D Lake $\overline{\mathbf{h}}_{\mathbf{s}}$ = 12.76 cm) to December 14$^{th}$ (Vee Lake $\overline{\mathbf{h}}_{\mathbf{s}}$ = 16.06 cm). During both field campaigns there was evidence of snow dunes present across the lakes. This study explored the distribution of snow over each lake (Figure 9), which showed local-scale variability of snow depths from redistribution of the snow across all the lakes (correlation lengths

between 6–19 m). We used semi-variogram analyses to determine the horizontal spacing of the snow dunes and found Long Lake to have the shortest correlation length (6.42 m). On Landing Lake, we saw an increase in correlation length throughout the winter season from ~7 m to ~19 m. The inferred variability length-scales are similarly supported in the literature, reporting correlation lengths from 5 to 20 m (Gunn et al., 2021a; Sturm and Liston, 2003)."

**Lines 361-369:** "The snow distribution over lake ice is known to be affected by wind and surrounding vegetation (Adams, 1976a). In this study we found weak relationships between the lake snow depth and distance to shoreline perimeter. On Finger Lake where we have more complete coverage of the lake, we found the snow depth to decline ~2 cm per meter from the shoreline to the centre of the lake but found no change on the additional three lakes. We believe this could be due to the lack of data representativeness around the shoreline and the difficulty associated with maneuvering the snowmobile in the deep, lighter snow at slow speeds, or the turbulent winds affecting which shoreline the snow will be distributed along. Winds reported at the Yellowknife weather station reached speeds above the ~14 to 39 km/h threshold required to transport snow (Li and Pomeroy, 1997), however, with the majority of strong winds coming from the northeast and northwest, our lack of data on the southern perimeter on each lake may also affect our findings."

- Line 17-19, "On average, the snow depth derived from GPR TWTs for the early winter season is estimated with a root mean square error (RMSE) of 1.58 cm and a mean bias error of -0.01 cm. For the late winter season on a deeper snowpack, the accuracy is estimated with RMSE of 2.86 cm and a mean bias error of 0.41 cm." Is the increase in mean bias error in the late winter season due to the effect of increased snow depth or the effect of deterioration?

  In comparing the difference in snow depth from March 2022 to December 2021, we found that there was only ever an increase in snow depth between the two dates and believe the increase in mean bias error is due to the effect of increased snow depth.

- Line 34-36, "As warming is occurring in Northern Canada at twice the global rate and is expected to continue to increase (Zhang et al, 2019)…" Has warming had an impact on snowfall? Is there a gradual increase or decrease in the amount of snow in winter?

  We have added to the manuscript to discuss how this years data could compare to previous years. This is done through the following text:

  **Lines 320-338: "**Lake freeze up for small lakes surrounding Yellowknife generally occurs during October, however, lake freeze up was reported to occur later this year compared to the 2018 to 2020 seasons based on Yellowknife's snowmobile association data. October air temperatures reported at the Yellowknife weather station showed a mean temperature increase of 4.4°C between 2020 (-1.85°C) and 2021 (2.6°C), and a 3.18°C increase when comparing to the 5-year and 10-year October mean air temperatures. Within the 2021 to 2022 water year, ~ 75 cm of snowfall was reported by the Yellowknife weather station, accounting for 46 % of total annual precipitation. In comparing the snowfall to previous

years, the 2021 to 2022 water year experienced 20% less snowfall than the 2020 to 2021 water year (~93 cm and 76% of total precipitation). In the past 5 to 10 years, on average, 40 to 45% more snowfall was reported compared to the 2021 to 2022 year. The timing and amount of snowfall will influence the lake ice composition, thickness, and phenology. Larger amounts of snow accumulation on thin, early season lake ice with reduced buoyancy will create leads and cause overflow, which increases the likelihood of snow ice growth. Thin and patchy snow ice (0 – 2 cm) was observed on the lake ice surface during the December and March field campaigns, making up 0% to 6% of the lake ice composition. Based on observations recorded up until March 2022, scarce amounts of snow ice were present, which suggests that minimal overflow occurred throughout the winter season on these four lakes prior to the beginning of ice break up.

- Line 75, "(2) validate the snow-depth retrieval algorithm using in situ observations…" Measuring uncompacted or compacted snow layers?

  Thank you for this question. The snow depth measurements were taken along side the track, so in the uncompacted snow – and that's why we use the uncompacted snow density. Although the depth and density are changing when the sled gets pulled over the snow, with no change in snow mass, the TWT will not be affected as clarified above.

- In addition to the spatial distribution of snow depth, I would like to know if you have also carried out research on the spatial distribution of ice thickness? Or is your technique actually focused on the identification of the snow-ice interface for shallow snow layers and is not actually an optimal technique for the identification of the ice-water interface?

  Thank you again for a great question. We collected snow depth and ice thickness simultaneously using the GPR. The GPR with the 1000 MHz  is capable of capturing the snow-ice and ice-water interface simultaneously, and the automated post-processing can pick both interfaces. This will be part of future research we are currently working on.

---

## Author Response (AR2)

**Comment on tc-2022-193**
**Response to Referee #1 Round 2**

Referee comment on "Mapping snow depth over lake ice in Canada's sub-arctic using ground-penetrating radar" by Alicia Pouw et al., Cryosphere Discussion, https://doi.org/10.5194/tc-2022-193, 2022

This MS has been revised according to the last round of review comments, and I have no further general comments. However, before considering publication, some details of expression need to be further improved. The following are my comments:

We are thankful to the reviewer for taking the time to review our revisions and provide additional valuable comments for improving the manuscript. Below, we provide the answers to the comments and questions raised by the reviewer with line numbers corresponding to the newly revised manuscript version. For convenience, comments from the reviewer are provided in black text. Responses to each comment are provided in blue text.

1) Abstract: Some main results derived from the observation data, such as the spatial distribution characteristics of snow cover, the relationship between snow cover on lake ice and snow cover on land, should be emphasized in the Abstract. At present, the main conclusions are mainly from the methodology.

Done - The abstract is modified.

**Lines 9-29**: Ice thickness across lake ice is influenced mainly by the presence of snow and its distribution, which affects the rate of lake ice growth. The distribution of snow depth over lake ice varies due to wind redistribution and snowpack metamorphism, which affect the variability of lake ice thickness. Measuring accurate and consistent snow depth data on lake ice is challenging and sparse to obtain. However, high spatial resolution lake snow depth observations are necessary for the next generation of thermodynamic lake ice models to improve the understanding of how the varying distribution of snow depth influences lake ice formation and growth. This study was conducted using ground-penetrating radar (GPR) acquisitions with ~9 cm sampling resolution along transects totaling ~44 km to map snow depth over four freshwater lakes in Canada's sub-arctic. The lake snow depth derived from GPR TWT resulted in an average relative error of under 10% when compared to in situ observations for the early and late winter season. The accuracy was assessed using 2,430 in situ snow depth observations. The snow depth derived from GPR TWTs for the early winter season was estimated with a root mean square error (RMSE) of 1.6 cm and a mean bias error of 0.01 cm, while the accuracy for the late winter season on a deeper snowpack was estimated with a RMSE of 2.9 cm and a mean bias error of 0.4 cm. The GPR-derived snow depths were interpolated to create 1 m spatial resolution snow depth maps. The findings showed improved lake snow depth retrieval accuracy and introduced a fast and efficient method to obtain high spatial resolution snow depth information. The results suggest that GPR acquisitions can be used to derive lake snow depth, providing a viable alternative to manual snow depth monitoring methods. The findings can lead to an improved understanding of snow and lake ice interactions, which is essential for northern communities' safety and wellbeing and the scientific modelling community.

2) Unit 10 (reference the revised MS with tracked change): "creating variability in the lake ice thickness"-- Snow depth and ice thickness are only physically dependent, but they cannot be

said to be created.

Thank you for this comment. We agree and have reworded to read as:

**Lines 10-11**: The distribution of snow depth over lake ice varies and due to wind redistribution and snowpack metamorphism, which affect the variability of lake ice thickness.

3) "a root mean square error (RMSE) of 1.58 cm", as well as the error of snow thickness in the full text, it is enough to retain one decimal place, for centimeter as the unit.

Throughout the manuscript we have adjusted the comparative statistics ($R^2$, RMSE, MAE, Bias) to one decimal place.

4) Unit 45 "As warming is occurring in Northern Canada": Whether the precipitation in the study region has a significant change trend?

Thank you for your comment. Precipitation changes have been reported an increase in precipitation with medium confidence due to the limited weather stations located in northern Canada, which is added to the revised version of the manuscript. We also discuss how the type of precipitation, whether it is snow or rain, can impact the formation of lake ice.

**Lines 58-68:** Snow and lake ice are sensitive to a change in daily air temperature (Rafat et al., 2023). Northern Canada is experiencing warming at twice the global rate, and it is expected that air temperature will continue to increase, along with precipitation (about 10%) in all seasons (Zhang et al., 2019). These changes can significantly impact the surface-atmosphere energy balance which can directly affect snow and lake ice conditions (Brown and Duguay, 2010). As a result of these changes, alterations in snow cover (Brown et al., 2021; Mudryk et al., 2017), snowfall (Vincent et al. 2018), lake ice phenology (timing of ice formation and break-up; Magnuson et al., 2000; Lei et al., 2012; Benson et al., 2011), and ice thickness and composition (Kholoptsev et al., 2021) are being observed. Spatial and temporal observations of lake snow and ice can provide insights to changes in climatic variables. Later freeze up and earlier break-up of ice cover lead to an extended open-season, which can affect lake surface water temperatures (i.e., Woolway et al., 2021), affecting the influencing lake biogeochemical processes (e.g., Adrian et al., 2009; Jeppesen et al., 2014).

**Lines 35-44:** While snowfall can accelerate the onset of lake freeze-up, once the ice has formed, the accumulation of snow hinders the ice growth in the water column (Adams, 1976a). Snow present on top of lake ice acts as an insulative barrier due to its lower thermal conductivity compared to ice. This process slows the growth rate of congelation ice (or black ice; Brown and Duguay, 2010; Leppäranta, 2015) and affects the heat released from the water column to the atmosphere. However, snow on lake ice can also impact the timing of melt and the ice-free season. The albedo of the snow surface reflects incoming solar radiation and can lead to a longer ice-on season (Jensen et al., 2007; Brown and Duguay, 2011; Robinson et al., 2021). Moreover, snow can produce ice growth as snow ice (or white ice), if the snow on the ice surface encounters water, forming slush, and refreezes (Leppäranta, 1983). This process can occur through the upwelling of water through leads, precipitation falling as rain, or heavy snow causing the depression of ice below the water level.

5) Unit 85 "when the distance between measuring locations is kept relatively small (<10 m)"--How did you get this "10 m"? In fact, the snow probe is very suitable for field observation in the range of hundreds of meters.

Thank you for this comment. We did not mean that the magnaprobe is not suitable for observations on the scale of hundreds of meters. In referencing less than 10 m, we are referring to the increased efficiency and ease to data collection when the sampling resolution is smaller. The efficiency in data collection using the magnaprobe decreases as the sampling area increases, even if the sampling resolution is increased. We have slightly adjusted the wording as follows:

**Line 93-96:** An advantage of using a magnaprobe is the increase in speed with which a depth and position measurement can be obtained compared to measuring with a traditional ruler and writing down the results. The highest boost in snow depth measurement efficiency occurs when the distance between measuring locations is kept relatively small.

6) The unit of wind speed is preferably m/s.

We have converted wind speeds to m/s instead of km/h.

**Line 138-142:** The most predominant winds in December and November came from the east (~27%) and had an average wind speed of 2.5 m/s, with the strongest winds coming from the northeast (~15%) reaching 9 m/s. Throughout January to March, the strongest winds came from the northwest (~22%) reaching 10 m/s, but frequent winds came from the northeast in January (~ 22%), northwest in February (~26%) and northeast, east, and northwest in march (~21%) travelling at 3 m/s on average, while very little winds were recorded from the south (~6%) between October to March.

**Line 412-415:** Winds reported at the Yellowknife weather station reached speeds above the ~4 to 11 m/s threshold required to transport snow (Li and Pomeroy, 1997), however, with the majority of strong winds coming from the northeast and northwest, our lack of data on the southern perimeter on each lake may also affect our findings.

7) Caption of Figure: you miss the panel a.

Thank you for noticing this missing detail. We have now added reference to the figure panel.

**Line 149-152:** Figure 1: This study focuses on (a) four lakes located north of Yellowknife, NWT, Canada, (b) Landing Lake, Finger Lake, Vee Lake, and Long Lake, shown on different scales depicting the area data collection took place (shaded colour). (c) The location of the GPR transects (Left) and in situ snow depth and density measurements (Right) on Vee Lake. (Background imagery: ESRI 2022, Landcover source: CCRS and NRCan, 2020)

8) Unit 230 "Landing-D = 190 kg/m3"--How is the density of snow obtained? When the snow is thick, whether the texture of snow is considered and the density of snow is measured at each sub-layer?

During data collection we obtained the bulk density of the snowpack at 6 to 10 locations per lake. Unfortunately at the time we did not excavate a snow pit and use the density cutters or

obtain the stratigraphy. Therefore, our observations are just based of an average bulk density for each lake. We have added more information in section 3.2 on how density observations were obtained:

**Lines 192-195:** For each lake, the bulk snow density was sampled at 6 to 10 locations through measuring the specific snow volume and weight of the vertical snow profile using a 5 cm diameter snow tube and an electronic scale with a 1 g accuracy. The bulk snow densities measured on each lake were averaged (Table 2) and used as a guide in determining the appropriate density to use for deriving the snow depth.

9) Figure 7: From the data in the figure, the error is different with different snow depth, and the thin snow corresponds to a relatively large error. This should be taken seriously.

We agree and in the manuscript we have addressed there is a threshold and the accuracy of the snow depth derived from the GPR-TWT decreases with thin snow with this method. The thinner snow depths (< 10 cm) are difficult to discern due to the imaging resolution associated with the 1000 MHz GPR sensor, as well as the associated errors in picking the travel times (~ 0.3 ns). We have added the following lines within the results and discussion sections in the revised manuscript to ensure that the limitation is clearly highlighted.

**Lines 292-293:** The snow depth derived from the GPR-TWT was consistently overestimated when compared to in situ observation for shallower snowpacks (<10 cm) across all four lakes.

**Lines 418-424**: This study revealed a limitation in deriving snow depths below 7cm using the 1000 MHz sensor, identifying a threshold. This finding is consistent with previous studies (Pfaffhuber et al., 2017), and as such, the in situ observations below 7 cm were excluded from the validation analysis. During the March 2022 campaign, seldomly snow depth was observed below 25 cm, meaning the vertical imaging resolution of 6.5 cm for the 1000 MHz sensor did not limit our data acquisition.

10) Unit 295 "The snow depth on Finger Lake showed a decrease of ~2 cm per meter as the distance from the perimeter"-- Whether the correlation length of snow depth will also change with the distance from the shoreline?

We found that the variability in snow depth is larger closer to the shoreline than further away on all four surveyed lakes. This would allow us to suspect the correlation lengths to be smaller closer to the shoreline than further away, however we do not quantify it as there was no significant trend shown across the lakes with snow depth and distances to shoreline.

**Lines 302-304:** The interpolated GPR-snow depths consistently show an increase in snow depth variability closer to the lake perimeter compared to areas farther from the shoreline and closer to the center of the lake.

11) Unit 365 "which suggests that minimal overflow"-- I don't understand the overflow here.

Thank you for this comment. The text has been modified in the revised version of the manuscript:

**Lines 364-372:** The timing and amount of snowfall influences the composition, thickness, and phenology of lake ice. Increased snow accumulation in early season and on thin lake ice that has reduced buoyancy will create leads and cause overflow (the upwelling of water from the water column), which increases the likelihood of snow ice growth. Thin and patchy snow ice (0–4 cm) was observed on the lake ice surface during the December and March field campaigns, comprising 0% to 6% of the lake ice composition. Based on observations up until March 2022, minimal amounts of snow ice were found, indicating that there was little overflow that occurred on these four lakes during the winter season prior to the beginning of ice break-up.

12) Unit 515 "however, the impact density variability has on lake ice formation needs to be further investigated"-- I also don't understand this sentence, maybe just remove it.

Thank you for this comment. We have removed the sentence.

13) 6 Conclusion: The expression of this section seems a bit messy, and I suggest further modification.

The conclusion is revised (**Lines 468–494**)

14) "The snow over lake ice has commonly been ignored when deriving lake ice thickness, with best practices for mapping ice thickness suggesting to avoid snow drifts and variable snowpacks, as it will estimate a thicker ice thickness due to the radar travel-time, but in reality, areas of snow drifts are expected to have a shallower ice thickness due to snow insulating the ice thickness and slowing ice growth. Variable snow depths are important areas across the lake to map for monitoring lake ice conditions as the ice thickness is expected to vary spatially. "-- These contents are not the conclusions derived from this study and have been introduced as background knowledge before, so it is suggested to delete these contents.

The paragraph has been deleted.

**Comment on tc-2022-193**
**Response to Referee #2 Round 2**

Referee comment on "Mapping snow depth over lake ice in Canada's sub-arctic using ground-penetrating radar" by Alicia Pouw et al., Cryosphere Discussion, https://doi.org/10.5194/tc-2022-193, 2022

I would like to thank the authors for their response and revisions. The manuscript has greatly improved and the majority of my previous comments have been satisfactorily addressed. However, two of my earlier comments have not been sufficiently addressed and must be in order for the manuscript to be publishable, in my opinion.

We are thankful to the reviewer for taking the time to review the manuscript a second time, and we appreciate their additional feedback. Below, we provide the answers to the comments and questions raised by the reviewer. For convenience, comments from this Reviewer are provided in black text. Responses to each comment are provided in blue text.

These are as follows:

1) The authors argue that the work is a novel advancement at implementing a new algorithm.

However, they did not develop the Wong et al. algorithm so I think the contribution is the matlab code. So, in order for this to be considered an advancement I would ask that the authors make this matlab script publicly available through something like CUAHSI Hydroshare or github.

Thank you for this comment. We have made revisions to the text to address this concern. Firstly, we have added additional details regarding the availability of the data and code section of the manuscript (**Lines 497-498**), stating that the code is available upon request. All post-processing steps are included within the workflow, including the signal processing, and prepared if and when requested. However, throughout the manuscript we do not claim to develop the Wong et al. algorithm and just state that we adapted the algorithm to make it usable for identifying the snow-ice and ice-water interface using a fully automated processing workflow (**Lines 118-124**). We focus on the derivation of relatively shallow snow depths using signal processing to remove the direct air wave that hides the snow-ice interface. We bring attention to the fact that it is possible to derive the lake snow depth and can be accomplished in a fast and efficient way.

**Lines 497-498:** All code used for data processing and analysis of this study are available from the corresponding author upon request.

**Lines 118-124:** We utilize extensive GPR two-way travel-time (TWT) observations and in situ observations of lake snow depth and density to complete the following objectives: (1) To improve the retrieval of lake-specific snow depth observations by applying a fully automated snow processing algorithm, (2) To validate the accuracy of the snow depth retrieval algorithm by comparing it to in situ observations, and (3) To spatially map the distribution of snow depth across lakes.

2) Additionally, I fundamentally disagree with the exclusion of values within the 6 m for comparison to your GPR technique. This implicitly assumes that your GPR processing is correct and is not actually testing your methods. Thus, the current error metrics are biased from the flawed methodology. I think that the authors must use all values within the 6 m window. I think it would be OK to discuss how this is a conservative estimate that could later be improved with a more accurate location for depth, etc. but I also suspect it will be within the 6.5 cm accuracy for the radar frequency as well.

Thank you for this follow-up comment. While we appreciate your perspective, we respectfully disagree. Throughout the manuscript, we explain our rational for using scenario 2 to determine the accuracy of our method for deriving snow depth over lake ice. Including all observations within the 6m radius would lead to an incorrect comparison of one observation to many (inverse distance weighted), when we know that the snow depth can vary on similar length scales across the lake. In our previous response we described how the snow depth variability within the 6m radius ranges from approximately 2 cm to 5 cm (standard deviation of snow depth), which could result in a variability of between 5% to 40% in the estimate when compared to the average snow depth. However, we have addressed this issue in the manuscript by including a section that explains and reports the error for the magnaprobe spatial location (Section 3.2 In situ Observations, **Lines 184-189**). We also suggest future work to improve the accuracy estimate of this method through enhancing the GPS data (Section 5. Discussion, **Lines 401-404**). In response to the reviewer's concern, we have included error metrics on the inclusion of all observations within the 6 m radius and our findings show a negligible increase of only ~1 cm in both the MAE and RMSE (**Lines 284-287**). These results demonstrate that there are minimal differences in the error statistics other than the $R^2$, which would be due to the variability in snow depths being on the same magnitude of the error in snow depth which, again, is confirmed with the correlation length scales (e.g. long lake correlation length = 6.42m with ~ 3.8cm standard deviation in snow depth at 6m).

**Lines 184-189:** The spatial accuracy for the magnaprobe GPS receiver with use in the Arctic has been reported as ± 5 to 10 m (Walker et al., 2020), with a 0.01 m depth precision (Sturm and Holmgren, 2018). With known limitations in the Magnaprobe GPS accuracy, we used the RTK GNSS rover to measure the location of 291 magnaprobe measurements spaced out along the sampling transects on three of the four lakes (Landing, Finger, Vee). We found the error from the

magnaprobe GPS to be between 1.72 m to 8.43 m, with a mean (± standard deviation) error of 4.44 ± 1 m.

**Lines 402-405:** The accuracy of this method may be improved by enhancing the spatial location of the in situ snow depth measurements and sampling more frequently within the length scales of each lake. In improving the spatial accuracy of the in situ snow depth observations and the frequency of measurements the accuracy of this method can be further assessed.

We have added additional lines stating the results of using all points within the 6m buffer:

**Lines 284-287:** Scenario 2 showed strong agreement between the in situ and estimated observations (Figure 7) with $R^2 = 0.63$, RMSE = 1.6 and MBE = 1.0 cm on average for all lakes. If considering all GPR-derived snow depth observations within the 6m radius there is minimal differences in the validation statistics (RMSE = 2.0 cm, MAE = 2.7 cm, Bias = 0.13 cm), however Scenario 2 is used for further analysis due to the variability in snow depth seen within the 6m radius (2.1 cm to 4.9 cm).